# Subcellular three-dimensional imaging deep through multicellular thick samples by structured illumination microscopy and adaptive optics

Ruizhe Lin[1], Edward T. Kipreos [2], Jie Zhu[3,4], Chang Hyun Khang[3] & Peter Kner [1✉]

Structured Illumination Microscopy enables live imaging with sub-diffraction resolution. Unfortunately, optical aberrations can lead to loss of resolution and artifacts in Structured Illumination Microscopy rendering the technique unusable in samples thicker than a single cell. Here we report on the combination of Adaptive Optics and Structured Illumination Microscopy enabling imaging with 150 nm lateral and 570 nm axial resolution at a depth of 80 μm through *Caenorhabditis elegans*. We demonstrate that Adaptive Optics improves the three-dimensional resolution, especially along the axial direction, and reduces artifacts, successfully realizing 3D-Structured Illumination Microscopy in a variety of biological samples.

[1] School of Electrical and Computer Engineering, University of Georgia, Athens, GA, USA. [2] Department of Cellular Biology, University of Georgia, Athens, GA, USA. [3] Department of Plant Biology, University of Georgia, Athens, GA, USA. [4]Present address: Department of Plant Pathology, University of California, Davis, CA, USA. ✉email: kner@engr.uga.edu

luorescence microscopy is a critical tool for biological dis-
covery, and recent advances in fluorescence microscopy
have led to significant advances in biology[1,2]. To obtain finer
detail in complex biological systems, different super-resolution
(SR) techniques have been developed to image below the dif-
fraction limit including structured illumination microscopy (SR-
SIM)[3,4], stimulated emission depletion microscopy (STED)[5,6],
photoactivated localization microscopy (PALM)[7,8], and stochastic
optical reconstruction microscopy (STORM)[9,10]. Among these SR
techniques, SIM stands out for its compatibility with live-cell
imaging[11]. As a widefield-based method, with low phototoxicity,
SIM achieves lateral resolutions of ~120 nm and axial resolutions
of ~300 nm[12,13], compared to ~250 nm lateral resolution in
conventional widefield microscopes. SIM requires significantly
fewer image acquisitions than required for STORM or PALM for
every SR image reconstruction, allowing for acquisition rates fast
enough for live cell imaging[4,11]. Moreover, three-dimensional
SIM (3D-SIM)[14] provides optical sectioning, which eliminates
out-of-focus light and enhances the image contrast. While STED
can also be applied to live imaging[15], STED typically requires
higher laser intensities and cannot image large fields of view as
rapidly.

Recently, there has been increased interest in in vivo SR ima-
ging in model organisms with higher spatial complexity such as
early-stage embryos and Caenorhabditis elegans (C. elegans)[16,17].
However, the application of SR-SIM has been limited to single
cells due to the degradation in image quality caused by optical
aberrations in thick tissues[18], which can be especially severe when
imaging deep into live organisms. As a consequence of the large
effective numerical aperture (NA), SR imaging methods are
generally more sensitive to optical aberrations[19]. SR-SIM relies on
an ideal and uniform point spread function (PSF) of the micro-
scope system, and the optical aberrations not only lead to noise,
artifacts, loss of image contrast, and resolution degradation in
SIM images[20] but can also prevent the SIM reconstruction
algorithm from finding a solution at all. Moreover, recent
research has found that even small optical aberrations, which
have minimal influence on the diffraction-limited image, will
cause severe artifacts in SIM[21,22]. Therefore, to successfully
implement 3D-SIM in thick tissues, it is critical to execute a
precise correction of the optical aberrations ahead of data
acquisition.

Aberrations can be classified into two categories, system
aberrations and sample-induced aberrations. System aberrations
are due to defects of optical elements or the imperfect optical
alignment in the microscope system and directly affect the PSF of
the microscope. Sample-induced aberrations are caused by the
optical properties of the biological sample, including the differ-
ence in refractive index between the sample and the surrounding
media, and the refractive index variations within the sample
itself[23,24]. Due to the diverse nature of biological samples, sample-
induced aberrations are case-specific, and, frequently, these
aberrations are spatially variant within the sample as well.

To correct the aberrations and restore the image quality,
adaptive optics (AO) provides a feasible solution[25] and has been
applied in several microscopy systems[26–29]. Direct wavefront
sensing employs a wavefront sensor to provide a direct instan-
taneous measurement of the wavefront[30]. Its fast response allows
the deformable mirror (DM) to be adjusted for every image frame
when imaging dynamic processes in live samples. Turcotte et al.[21]
reported a combination of 2D SR-SIM (increased lateral resolu-
tion and optical sectioning) and direct wavefront sensing AO to
image the dynamics of dendrites and dendritic spines in the living
mouse brain in vivo. Li et al.[22] reported an optical-sectioning SIM
incorporating direct wavefront sensing AO that achieved fast,
high-resolution in vivo structural and functional imaging of

neurons in live model animals. Although possessing good accu-
racy and high-speed, the direct wavefront sensing method
requires a dedicated wavefront sensing system (e.g., a
Shack–Hartmann wavefront sensor (SHWFS)) which increases
the system complexity. SHFWS works best with an isolated guide-
star (e.g., two photon-induced fluorescent guide star[31] or fluor-
escent protein guide-star[32]), making it challenging to combine
with widefield microscopy. Turcotte et al.[21] and Li et al.[22] both
used two-photon excitation to generate the "guide star" for
SHWFS, which dramatically increases the cost and system com-
plexity. Moreover, when the wavefront sensing system fails to
form an identifiable image of the guide-star due to large aberra-
tions or a highly scattering medium, the correction can fail. A
sensorless model-based AO method[33] is more resistant to scat-
tering and can be operated upon highly aberrated images. It relies
on a series of artificial aberration trials and an image quality
metric function. The optimization is conducted iteratively by
maximizing the metric value. This AO method has been applied
to optical sectioning SIM[34,35] and two-dimensional SR-SIM[20].
Žurauskas et al.[36] reported on a sensorless AO method that
ensures adequate sampling of high frequency information by
using a customized illumination pattern. The samples being
imaged are flat cells cultured and mounted on glass, which do not
present the large and complicated aberrations existing in thick
multicellular organisms. All the above-mentioned work combin-
ing AO and SIM does not increase the resolution beyond the
diffraction limit in the axial direction.

In this article, we demonstrate the application of AO to 3D-
SIM, achieving a resolution of 150 nm laterally and 570 nm axi-
ally (emission wavelength at 670 nm, NA of 1.2), along with
optical sectioning, compared to a resolution of 280 nm laterally
and 930 nm axially in widefield imaging. First, we demonstrate
that the sensorless AO method based on widefield images is
capable of correcting system aberrations and sample induced
aberrations in a variety of samples. We then show confocal spot
based sensorless AO that achieves good aberration correction
with enhanced robustness. We further use the confocal illumi-
nation to create a guide-star for direct wavefront sensing, showing
effective aberration correction and significant improvement in
3D-SIM imaging. To demonstrate AO-3DSIM, we take three-
dimensional images of α-TN4 cells, fluorescent beads under the
body of a C. elegans, the GFP-labeled endoplasmic reticulum (ER)
of rice blast fungus inside rice plant cells and GFP-labeled axons
and adherens junctions (AJs) in C. elegans. The final three-
dimensional images clearly reveal the object of interest in all three
dimensions, and the application of AO yields a significant
improvement in the image. Image signal to noise ratio (SNR) is
significantly increased with the application of AO, and
fine structures which are not identifiable when aberrations are
present, become well resolved and clear.

## Results

**The effect of optical aberrations on SIM.** Figure 1 shows
simulation results of the impact of three basic aberration modes
(astigmatism, coma, and spherical aberration) on the 3D-SIM
spatial images, and effective Optical transfer functions (OTFs). As
they do in the widefield image, the aberrations result in additional
intensity structure around the PSF which will affect the signal to
noise. The optical aberrations are mathematically described by an
expansion of the wavefront in the back pupil plane in Zernike
polynomials, which are a complete, orthogonal set of eigenfunc-
tions defined over the unit circle[37]. As is well known, the low-
order Zernike modes correspond to the typical aberrations of an
optical system[38]. Within a limited amplitude range, each of the
orthogonal Zernike modes affects the image independently from

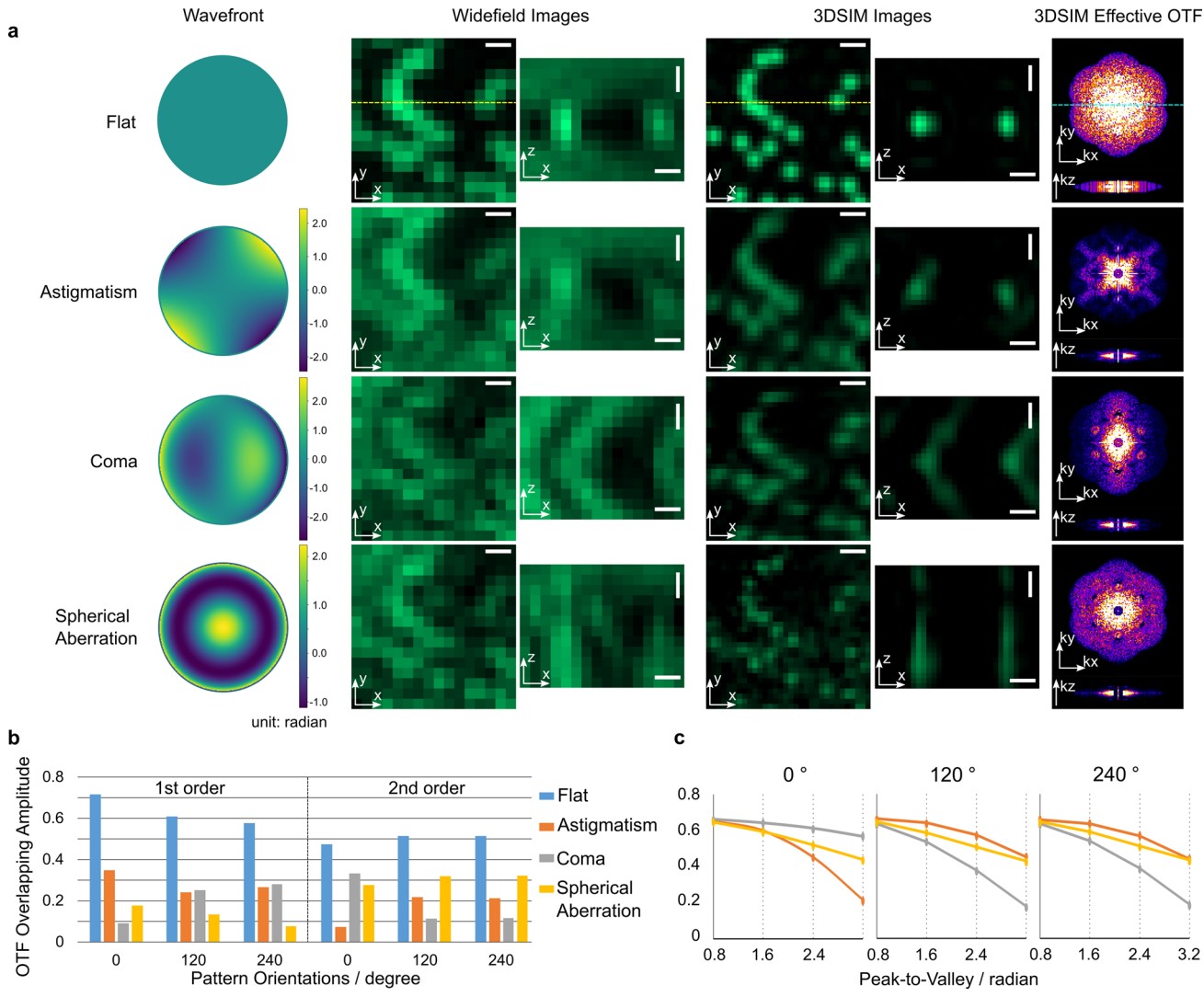

**Fig. 1 Simulations of the effect of aberrations on 3D-SIM. a** Four different wavefronts (flat, astigmatism, coma, spherical aberration) and their corresponding widefield images, 3D-SIM images, and effective OTFs of the 3D-SIM images. The x-z slices are cross sections through the dashed lines in their corresponding x-y slices. The images in the same column are on the same intensity scale. All horizontal scale bars are 0.25 μm. All vertical scale bars are 0.5 μm. **b** The OTF overlapping amplitudes of the first and second order frequency components computed from the images in **a** by our 3D-SIM reconstruction algorithm. **c** The OTF overlapping amplitudes of three pattern orientations (0°, 120°, 240°) for the three aberration modes (astigmatism, coma, spherical aberration) of increasing peak-to-valley wavefront errors. The OTF overlapping amplitude is the parameter in the reconstruction algorithm, which determines the pattern frequency and strength. The amplitude value reaches its maximum when the position of the shifted data, $\widetilde{D}_{j,m}$, matches the pattern's wave vector.

others. This linearity allows us to investigate the imaging with each Zernike mode separately. Unlike incoherent structured illumination microscopy[34], aberrations in the back pupil plane do not affect the strength of a coherent structured illumination pattern although they do affect the phase and spatial frequency. Therefore, in principle, the strength of the higher-order contributions to SIM should not be affected. However, SIM relies on the comparison of information in the overlap between $\widetilde{D}_0$ and $\widetilde{D}_{j,\pm1}$ (See Eq. 1). This overlap is strongly affected by the reduction in the OTF, $\widetilde{H}$. As we can see in the bar graphs in Fig. 1b, the OTF overlap amplitudes drop significantly with aberrations. Furthermore, the OTF overlap amplitude will be strongly affected by the orientation of the pattern with respect to the aberration. This can be seen in the graphs in Fig. 1c, where we calculate the overlap amplitudes for different pattern orientations and aberration amplitudes. For the results in Fig. 1, we ran the simulations

with preset pattern frequency and phase, so we achieve the optimal parameter fitting when running the reconstruction algorithm on aberrated images. However, in real cases, the drop of pattern amplitudes could result in inaccurate computation of the pattern frequency and phase, which would introduce even more serious artifacts than what we see in Fig. 1.

**The AO-3DSIM system.** We designed an optical microscopy system that can be switched among three illumination schemes (widefield illumination, structured illumination, and confocal illumination) and can operate with both senorless and direct wavefront sensing AO. As shown in Fig. 2, three rotatable mirrors (RMa, RMb, and RMc) are used to control the excitation and emission light paths for setup switching. The structured illumination is generated by a pattern generating unit acting as a binary phase grating (inset, Fig. 2)[11]. Pinhole 1 is placed at a conjugate

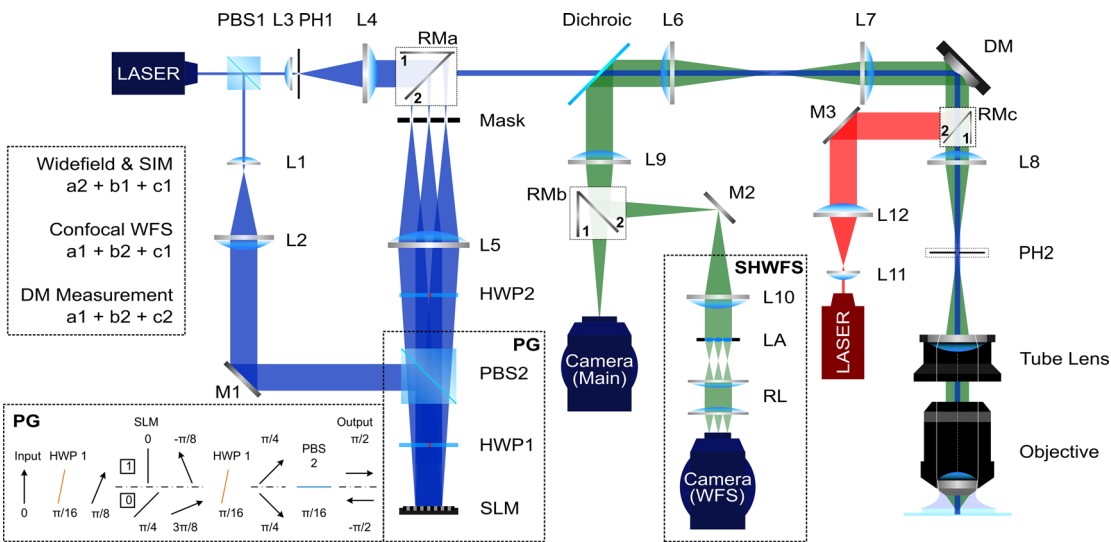

**Fig. 2 Schematic of the AO-3DSIM microscopy system.** RMa, RMb, and RMc are rotatable mirrors that can be switched between position 1 and 2. They are used for switching the light beam between optical setups, the position combinations are listed in the dashed line box on the left. PG pattern generation. The polarization direction indicated by arrows is shown after each element for both on (1) and off (0) pixels. SHWFS Shack–Hartmann wavefront sensor, SLM spatial light modulator, HWP half-wave plate, PBS polarized beam splitter, DM deformable mirror, PH1-PH2 pinhole, LA lenslet array, RL relay lens, M1-M3 flat mirror, L1-L12 lens. $f_1 = 40$ mm, $f_2 = 200$ mm, $f_3 = 17$ mm, $f_4 = 300$ mm, $f_5 = 250/300$ mm, $f_6 = 100.7$ mm, $f_7 = 250$ mm, $f_8 = 250$ mm, $f_9 = 300$ mm, $f_{10} = 170$ mm, $f_{11} = 20$ mm, $f_{12} = 250$ mm. The blue represents the excitation path (488 nm); the green path is the emission; and the red path is used to calibrate the DM.

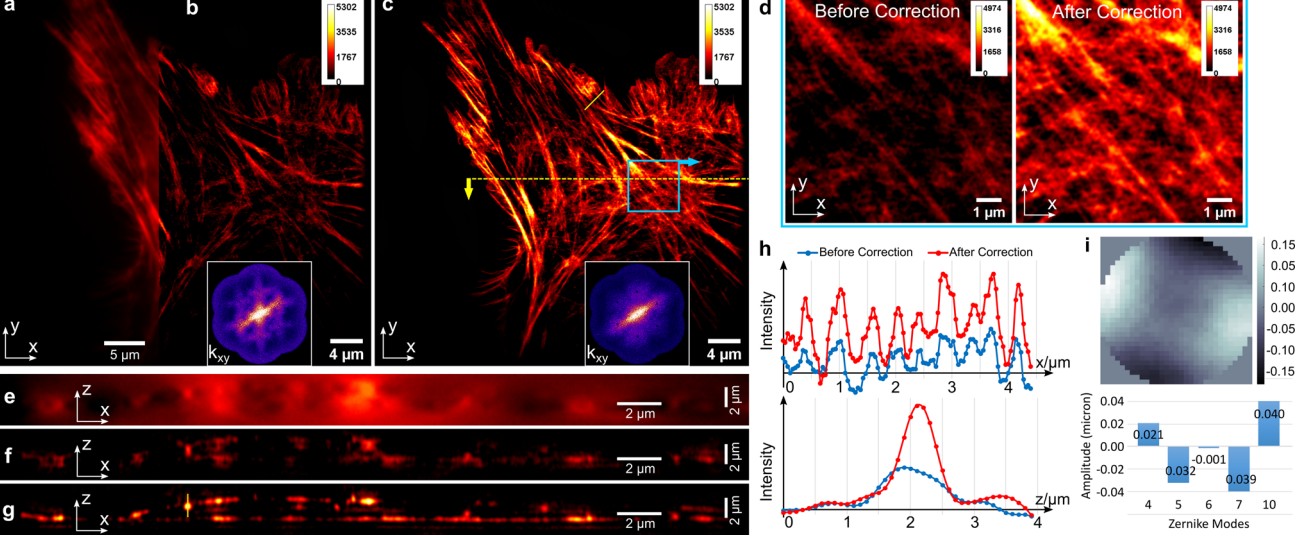

**Fig. 3 The actin of α-TN4 lens epithelial cell, labeled with Phalloidin-iFluor 647. a** Widefield fluorescence image with AO correction. **b** 3D-SIM image without AO correction and the OTF section at $k_z = 0$ (inset). **c** 3D-SIM image with AO correction and the OTF section at $k_z = 0$ (inset). All three images are x-y slice of the 3D images in the focal plane (z = 0). **d** Zoomed-in views of the actin structures before (left) and after (right) AO correction. The x-z slice cut through the dashed yellow line in **c**. **e** widefield with AO; **f** 3D-SIM; **g** AO-3DSIM. **h** Intensity profile along the solid yellow lines in x-y and x-z slice images, respectively. **i** The amplitudes of Zernike modes and the wavefront posted on the DM, units in microns.

image plane to ensure a diffraction limited focal spot for confocal illumination. Pinhole 2 is placed at the first image plane to block the out-of-focus light when running the SHWFS with confocal illumination. We implemented 3D-SIM as developed by Gustafsson et al.[13] and applied image-based sensorless AO, confocal sensorless AO, and direct wavefront sensing AO methods for sample-induced aberration correction.

**Image-based sensorless AO-3DSIM.** We first applied sensorless AO on tissue culture cells (α-TN4 lens epithelial cells on a #1.5 glass coverslip) to demonstrate the correction of system

aberrations. The imaging was initiated with all the DM actuators set to 0. The size of the 3D image stack is $45.5\,\mu m \times 45.5\,\mu m \times 5.2\,\mu m$ in the $x, y,$ and $z$ dimensions, respectively. The comparison between Fig. 3a and b shows the resolution enhancement from widefield to 3D-SIM. And the comparison between Fig. 3b and c shows the effect of AO correction on the 3D-SIM image. The Fourier transform (FT) of the 3D-SIM image (insets) illustrates the aberrations clearly in the form of a cross shape in each OTF copy, revealing a dominating aberration of astigmatism (Zernike mode 5, Noll ordering[39]). Because the system aberrations are isoplanatic, or nearly so, we

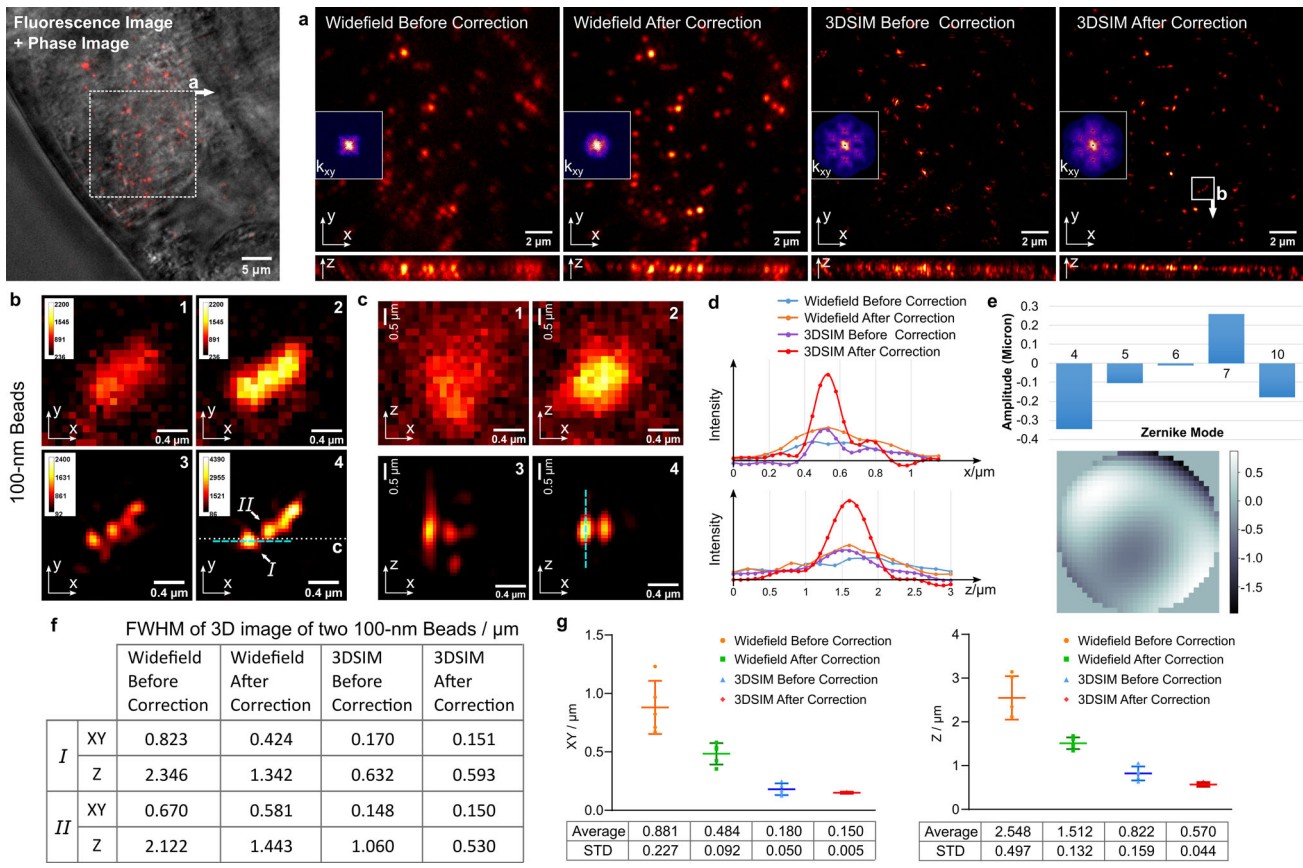

**Fig. 4 Mixture of 100-nm and 200-nm fluorescent beads under *C. elegans*. a** In-focus x-y section images of area within white dotted-line square and the maximum intensity projection of their x-z sections, insets are the image spectrum. **b** Zoomed-in views of the area within the white square. **c** The x-z slice cut through the white dashed line. **d** Intensity profiles plotted along the blue dashed lines along both the x and z axis in widefield and 3D-SIM images (**1**. Widefield image without AO correction; **2**. Widefield image with AO correction; **3**. 3D-SIM image without AO correction; **4**. 3D-SIM image with AO correction). **e** The amplitudes of Zernike modes and the corrective wavefront applied on the DM, units in microns. **f** The table of the FWHM of 3D images of two 100-nm beads. **g** The scatter diagram of the average values and standard deviations of the FWHM of 3D images of 100-nm beads. $n = 5$ independent beads were measured. Source data are provided as a Source Data file.

conducted the AO correction directly on the whole field of view. A bargraph of the applied Zernike modes and the resulting wavefront are shown in Fig. 3i. As shown in the AO-3DSIM image (Fig. 3c) and its corresponding FT, with AO correction, more fine features are evident in the spatial image and the cross shape disappears from all OTF copies. The improvement in the axial direction can be seen by a comparison of the x-z slices in Fig. 3e (widefield), Fig. 3f (3D-SIM), and Fig. 3g (AO-3DSIM). The out-of-focus light, which overwhelms the widefield image, is removed by 3D-SIM. And AO correction improves the signal to noise and the fidelity of the image. The image contrast gets higher from the widefield image without AO to the 3D-SIM image with AO in both the lateral and axial directions. As shown in Fig. 3h, by comparing the intensity profiles along the x axis of 3D-SIM images before and after AO correction, we see that the signal intensity is doubled. From the intensity profiles along the z axis, we see the full width half maximum (FWHM) gets smaller, reducing from 0.828 to 0.554 µm, and the peak intensity increases by 1.25x.

We then imaged 100-nm fluorescent beads under an adult *C. elegans* hermaphrodite to demonstrate the correction of sample-induced aberrations. The adult *C. elegans* has a length of ~1 mm and a diameter of ~80 µm. Figure 4a, b illustrate the improvement in 3D resolution and the removal of out-of-focus blur with the application of AO and 3D-SIM. Here the imaging is initiated with

the DM set to correct system aberrations. Similar to Fig. 3, we can also identify the AO correction from the disappearance of the cross shape in the frequency spectrum shown in Fig. 4a (inset); here, the dominant astigmatism mode is Zernike mode 4, now due to the body of the worm. Figure 4c, d give a more detailed view of the advantages of AO-3DSIM through a close-up look at four 100-nm beads. Without AO correction, we can see artifacts around the beads in all three dimensions. The image is distorted in the lateral plane and elongated along the z-axis. After AO correction, the shape of each bead is recovered, and the peak intensity is increased. We see significantly lower noise around the beads, and the intensity distribution along the z-axis is more confined to the focal plane. A quantitative comparison can be seen from the intensity profiles in the x and z directions in Fig. 4d. By Gaussian fitting of the intensity profiles in the x and z directions, and calculating the FWHM, as shown in Fig. 3f, g, we get a quantitative evaluation of the improvement due to AO and SIM. More quantitative results can be seen in Supplementary Fig. S2. We conclude that the AO-3DSIM system achieves a resolution of ~150 nm laterally and ~570 nm axially.

Then we demonstrated the correction of sample-induced aberrations by sensorless AO on the ER of the filamentous fungus *Magnaporthe oryzae*, also known as rice blast fungus, growing inside rice plant cells. The fungus constitutively expresses GFP fused with an N-terminal secretory signal peptide

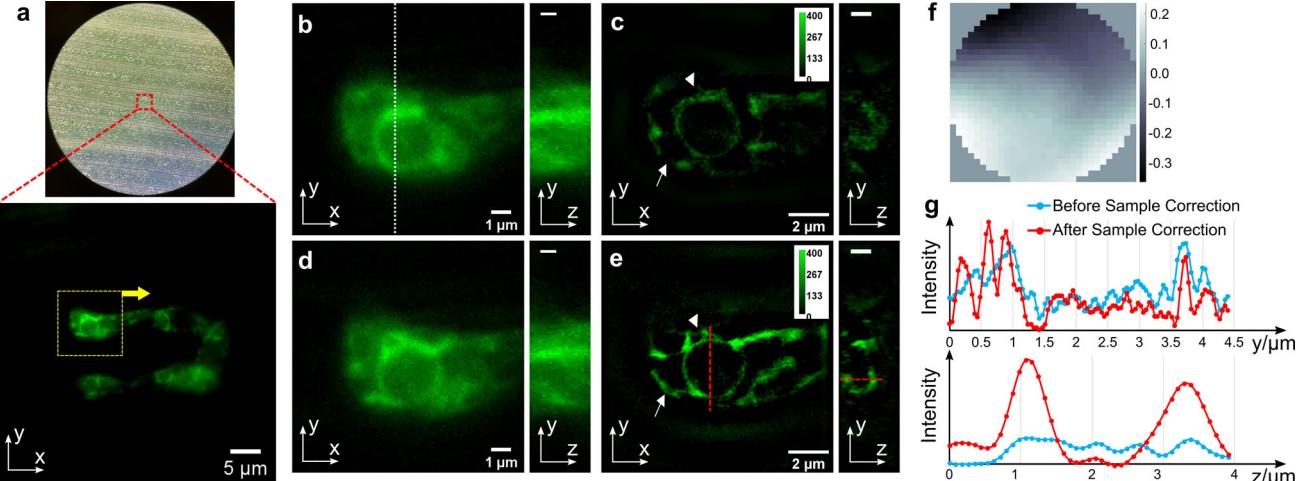

**Fig. 5 GFP-labeled endoplasmic reticulum of live *M. oryzae* hyphal cells growing inside rice plant cells. a** Brightfield image of the rice sheath tissue and the fluorescence image of the GFP-labeled ER of hyphal cells. The yellow dashed-line square marks the area of interest in sample aberration correction. **b** Widefield image with only system aberration corrected. **c** 3D-SIM image with only system aberration corrected. **d** Widefield image with both sample and system aberrations corrected. **e** 3D-SIM image with both sample and system aberrations corrected. All x-y slices are in focus and the y-z slices are cross sections cut through the white dotted lines. **f** The corrective wavefront applied to the DM, units in microns. **g** Intensity profile plotted across the red dashed lines along both the y and z axis in 3D-SIM images. Arrowheads indicate perinuclear ER and arrows indicate peripheral ER.

and a C-terminal ER retention signal peptide, which labels the ER network[40]. For live-cell imaging, we used hand-cut optically clear rice sheath tissue, consisting of a layer of epidermal cells, within which the fungus resides, and a few underlying layers of mesophyll cells (Fig. 5a). The sample is ~60 μm thick, and we are imaging 30 h after inoculation when the fungus is ~20 μm below the sample surface. Due to the spatial variance over the whole field of view, we conducted AO correction upon the area inside the yellow dotted square as shown in Fig. 5a. In filamentous fungi, the ER can be seen as the spherical perinuclear ER and the reticulate peripheral ER along the hypha at the fluorescence light microscopy level[41]. Figure 5e clearly shows smoother and more continuous perinuclear ER (arrowheads) and peripheral ER (arrows), which appear wider in Fig. 5c, g. Also, there is substantial improvement in the axial resolution after AO correction. The z slice image in Fig. 5c, e shows a ring-shaped EGFP fluorescence pattern, demonstrating the spherical organization of perinuclear ER, which is not resolved in Fig. 5c. The intensity profile plots in Fig. 5g show the intensity increasing and FWHM narrowing after AO correction in both the lateral and axial directions. Since the correction is conducted within the square area with dashed yellow lines, the correction inside the area of interest is better than that in other areas. Although a general improvement can be seen over the entire image, the optimal performance is still localized within the correction area.

Next, we applied the sensorless AO method on *C. elegans* modified to express GFP fused to the protein tyramine beta-hydroxylase (TBH-1) in the RIC neurons[42]. TBH-1 is an enzyme involved in the octopamine biosynthetic process. In the absence of food, octopamine is released from the RIC neurons and activates the octopamine receptor SER-3 in the cholinergic SIA neurons. The structural details of the RICR/L neurons are illustrated in the graphic rendition shown in Fig. 6a. We took a 4 μm stack along the z-axis with a step of 0.2 μm for 3D-SIM, focusing on the axons before the ring as the image overlay in Fig. 6a shows. The effect of AO correction can be seen by comparing Fig. 6b, c in widefield and Fig. 6d, e in 3D-SIM. As can be seen in Fig. 6a, there are two RIC interneurons that travel towards the posterior of the worm from the pharynx. In Fig. 6d, there appear to be three filaments along the x-direction as

indicated by three white arrow heads. After AO correction, Fig. 6e, it is clear that there are only two fibers, as indicated by two green arrow heads. In the y-z slices showing the fiber cross sections (insets), the peak intensity after AO correction is higher than before AO correction, while the noise level is much lower. The isolated nerve fibers can be clearly recognized as continuous in the AO corrected image.

**Confocal sensorless AO-3DSIM.** For images with unidentifiable structural features and low SNR, the sensorless AO method based on widefield images loses its efficiency due to inaccurate image quality assessment. For example, in Fig. 7 we image the area around the anterior pharyngeal bulb in *C. elegans* expressing the *ajm-1*::GFP reporter[43]. AJs are membrane-anchored structures that stabilize cellular contacts. *C. elegans* is a model animal for studying the formation and regulation of adhesive structures in vivo[44]. The pharynx of *C. elegans* is comprised of striated muscle with an anterior bulb that acts to pump bacteria into the digestive tract and a posterior bulb that crushes the bacteria as the animals feed[45]. The AJs in Fig. 7 are located within the anterior bulb of the pharynx[46] where they presumably function to maintain structural integrity during pharyngeal pumping. As AJs are embedded within the anterior bulb and surrounded by multiple striated muscle cells, which are highly inhomogeneous, the GFP signals are weak and smeared out due to the aberrations and strong scattering. To enhance the robustness of the sensorless AO method in this case, we used confocal illumination to form a focal spot on the sample and conducted sensorless AO based on the image of the confocal spot (inset, Fig. 7a). The confocal spot serves to enhance the high frequency content of the image for correction and target the correction to the area of interest. We can see the effect of AO by comparing the 3D-SIM images of the two AJs in Fig. 7b, c (inside the blue square) and d (inside the orange square) before (left) and after (right) the sample aberration correction. The improvement with AO can be also seen by comparing the effective OTF of the 3D-SIM images before (upper) and after (bottom) sample aberration correction in Fig. 7e. We can see a significant change in the distribution and intensity of the frequency components, especially in the high

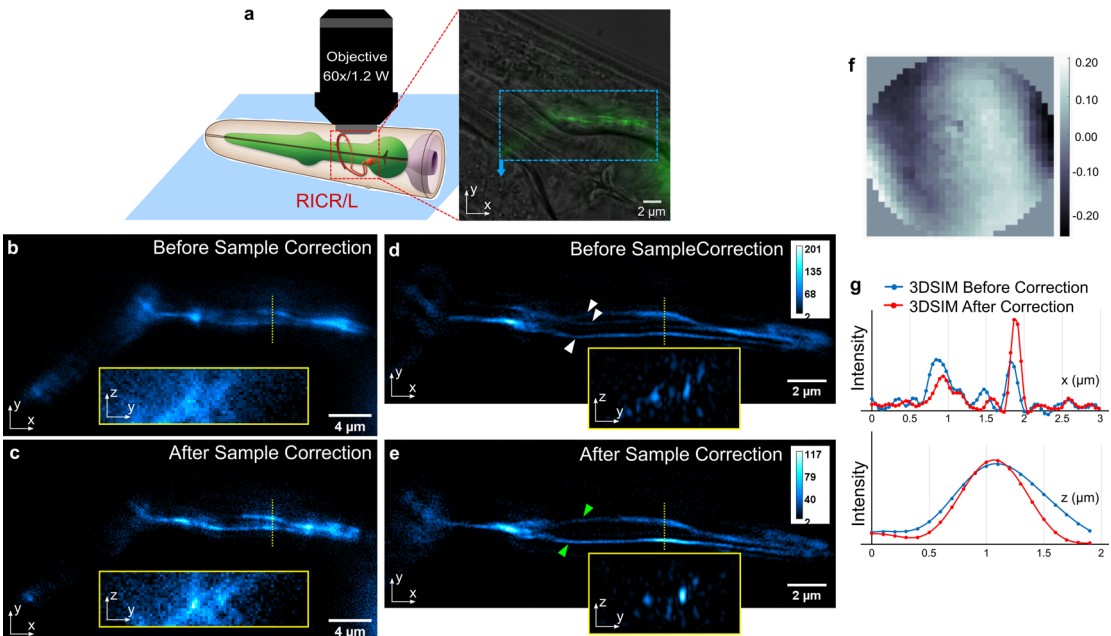

**Fig. 6 Live *C. elegans* with TBH-1::GFP expressed in RIC interneurons in the lateral ganglion. a** Graphic rendition of the RICR/L neurons in adult *C. elegans* (graphic from nervous system poster in WormAtlas[60]: http://www.wormatlas.org) and the overlay of widefield fluorescence image and DIC phase image. **b** Widefield image of axon before AO correction. **c** Widefield image of axon after AO correction. **d** 3D-SIM image of axon before AO correction and the y-z slice cut through the dashed yellow line. **e** 3D-SIM image of axon after AO correction and the y-z slice cut through the dashed yellow line. **f** The corrective wavefront applied on the DM, units in microns. **g** The line profiles across the axon in 3D-SIM images.

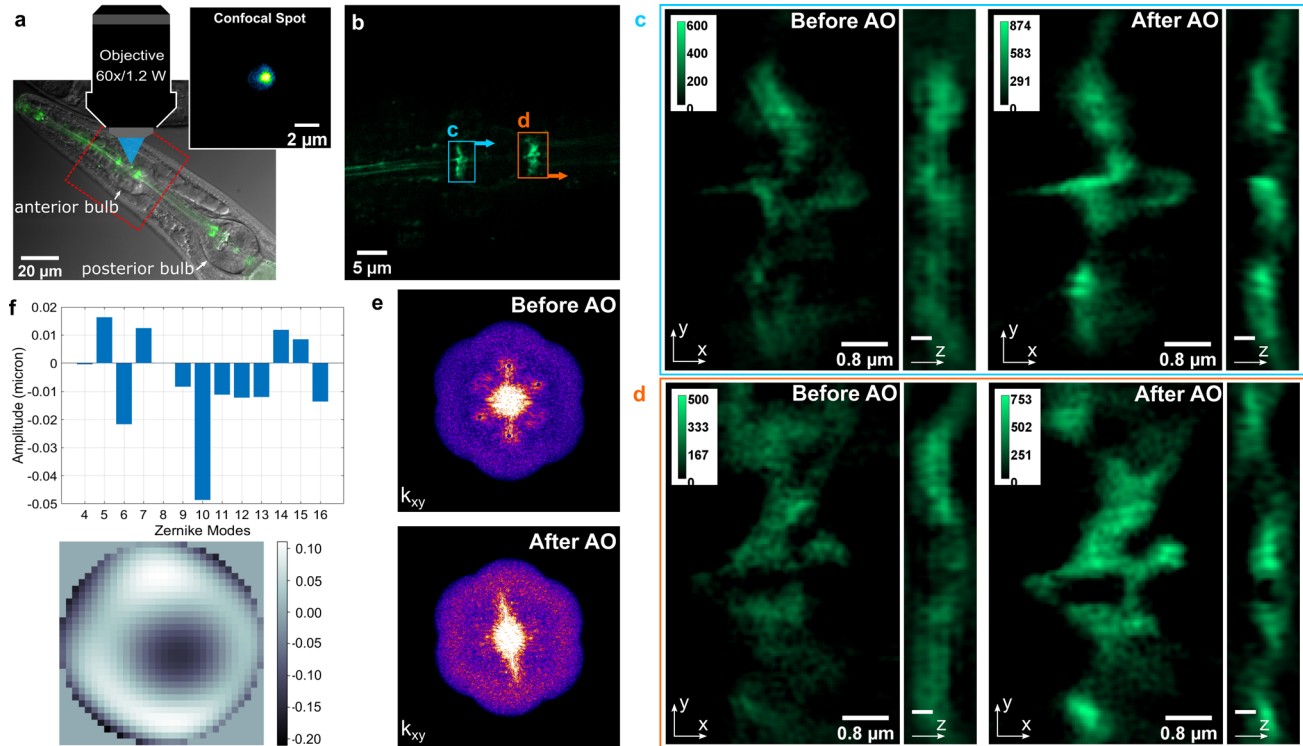

**Fig. 7 Live *C. elegans* expressing the adherens junctions marker *ajm-1*::GFP. a** An image overlay of a fluorescence image and a DIC phase image to illustrate the location of the AJs inside *C. elegans* and the image of the confocal spot (inset). **b** Widefield image of AJs in the anterior bulb of the pharynx. **c** 3D-SIM images of AJs in area *a* before and after AO correction. **d** 3D-SIM images of AJs in area *b* before and after AO correction. **e** Effective OTF of 3D-SIM images before (upper) and after (bottom) sample aberration correction. **f** The amplitudes of Zernike modes and the applied wavefront.

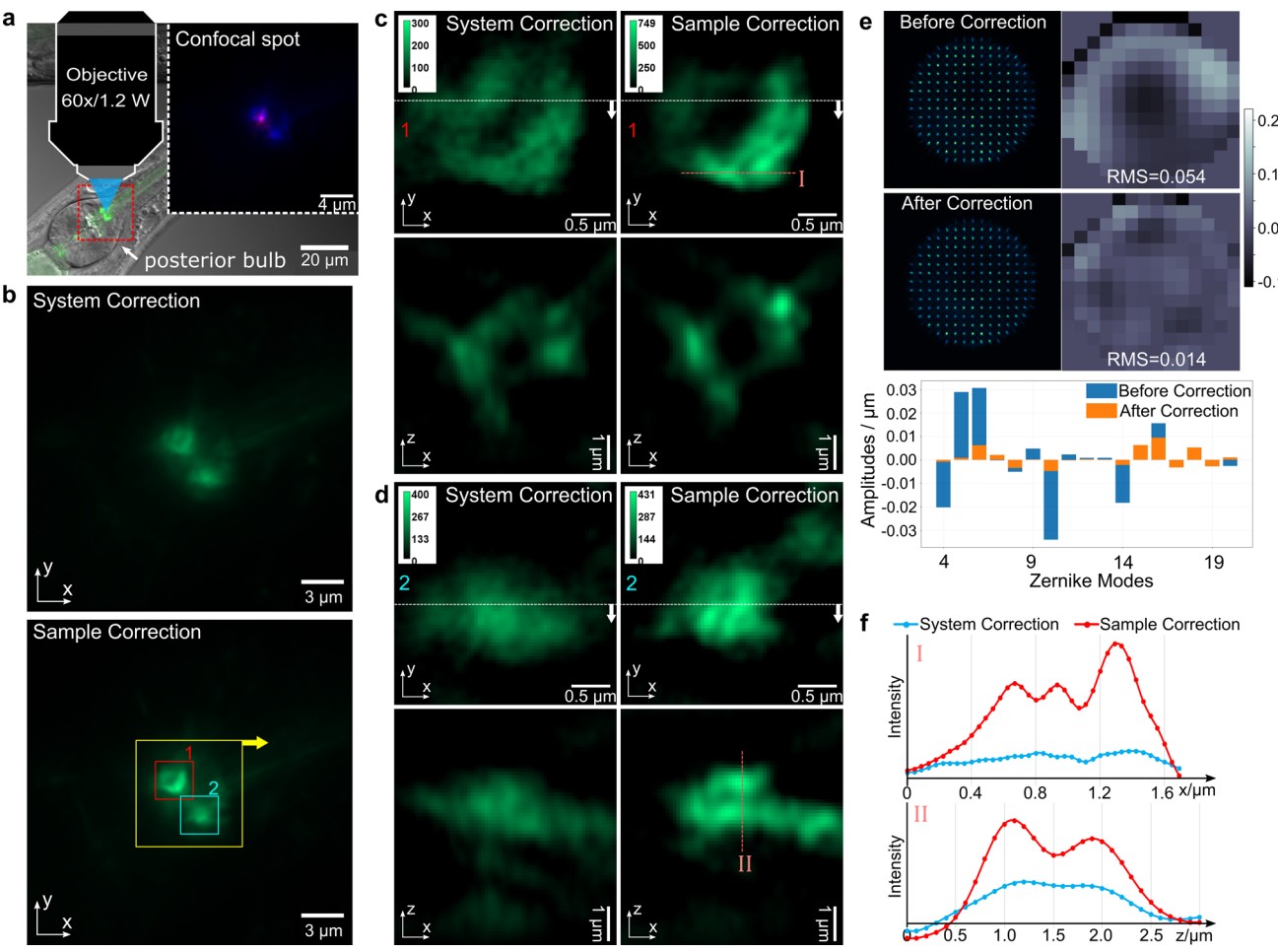

**Fig. 8 Live *C. elegans* expressing the adherens junctions marker *ajm-1*::GFP. a** The image overlay of a fluorescence image and a DIC phase image to illustrate the location of the AJs in the posterior bulb of the pharynx and the image of the fluorescent confocal spot (inset). **b** Widefield images of AJs in the posterior bulb of the pharynx before and after sample correction. **c** 3D-SIM image of AJ 1 (inside red square) before and after sample correction, including the in-focus x-y slices and x-z slices cut through the dotted white lines. **d** 3D-SIM image of AJ 2 (inside blue square) before and after sample correction, including the in-focus x-y slices and x-z slices cut through the dotted white lines. The images of the same area are set in the same intensity scale. **e** Focal spot array captured on the SHWFS, the measured wavefront before and after sample correction, and the Zernike mode decompositions of the measured wavefront. Units in microns. **f** Intensity profile plotted across the red dashed lines along both the x and z axis in 3D-SIM images.

frequency bands, revealing that AO correction restores information, improves image quality, and, more importantly, makes SIM work properly.

**Confocal direct wavefront sensing AO-3DSIM.** Operating sensorless AO on live *C. elegans* requires the worm to stay still during the iterations of optimization which require at least 25 image acquisitions, or more if higher order Zernike modes are included. The worm movement during that period will degrade the AO correction performance due to changes in the confocal spot intensity. Therefore, in order to achieve a faster and more accurate aberration correction, we applied direct wavefront sensing to *C. elegans*, in order to measure the wavefront distortion with a single image acquisition. We used the confocal spot, as shown in the Fig. 8a (inset, the purple dot), as the guide star for the SHWFS. To demonstrate the direct wavefront sensing method, we imaged bundles of AJs inside the posterior bulb, as shown in Fig. 8a. In the 3D-SIM image after sample correction, Fig. 8c, these bundles appear sharper. The overall signal intensity increases while the noise level and artifacts are suppressed. The wavefront decomposition in Fig. 8e shows that direct wavefront sensing successfully reduces those Zernike

modes with high amplitudes, such as Zernike mode 4, 5, 6, 10, and 14.

We further applied direct wavefront sensing to the GFP-labeled ER of the rice blast fungus growing inside rice plant cells. In Fig. 9, we are imaging 48 h post inoculation, and the fungus is ~30 μm below the sample surface. The thickness of hand-cut rice sheath tissue, the profile shape of the epidermal cells, and the depth of the fungal colonization jointly result in blurry images with low SNR as shown in Fig. 9a. The effect of the direct wavefront sensing AO method is evident in the improvement of the SNR, image sharpness, and structural definition, showing the correct morphology of the hollow structure and the ER of the hypha. The hollow structure is presumed to be a vacuole (Fig. 9b) or a nucleus (Fig. 9c), which should have no fluorescence in this fungal strain. as we can see from the comparison of 3D-SIM images before and after, sample aberration correction clearly reveals the exclusion of the GFP-labeled ER from the hollow structure in all three dimensions (in Fig. 9b and c). A significant image quality improvement in both lateral and axial dimensions can be observed from the spatial images and the intensity profiles plotted in Fig. 9d as well. As the wavefront sensing results show in Fig. 9e, f, the sample induces strong spherical aberration along with astigmatism and coma, yielding a wavefront with peak-to-

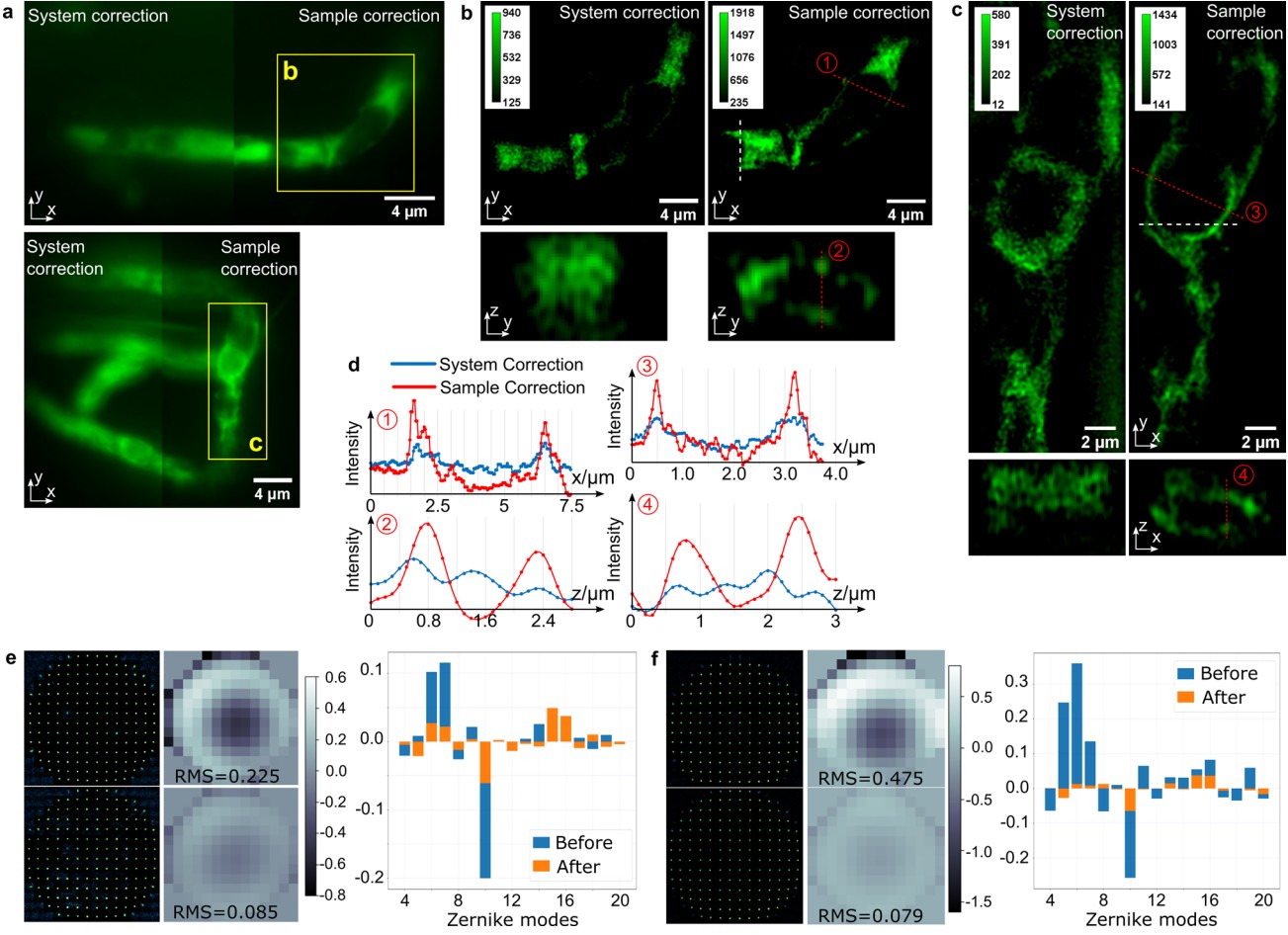

**Fig. 9 GFP-labeled endoplasmic reticulum of live *M. oryzae* hyphal cells growing inside rice plant cells. a** Widefield images of ER before (left) and after (right) sample aberration correction. **b** 3D-SIM image of ER inside square area b before (left) and after (right) sample aberration correction and the cross section cut through the white dashed line. **c** 3D-SIM image of ER inside square area c before (left) and after (right) sample aberration correction and the cross section cut through the white dashed line. **d** Intensity profiles plotted across the four red dashed lines. **e** The measured wavefronts of area b, and their Zernike mode decompositions before and after sample aberration correction (units in microns). **f** The measured wavefronts of area c and their Zernike modes decompositions before and after sample aberration correction (units in microns).

valley (PV) of ~2 µm and standard deviations (STD) of 0.22 µm and 0.46 µm for Fig. 9b, c, respectively. After sample aberration correction, the wavefront PV value drops to less than 0.3 µm and the STD value drops to ~0.07 µm. The Zernike decomposition clearly shows the suppression of previously strong Zernike modes, 5, 6, and 10. Comparing to the result in Fig. 5, we find the direct wavefront sensing method does a better job correcting the spherical aberration than the sensorless method does.

Finally, we applied direct wavefront sensing to the GFP-labeled RIC neuron inside the *C. elegans* (strain MT9971). From the results shown in Fig. 10, we can see the improvement by the direct wavefront sensing AO method in 3D-SIM over a 2 µm axial range near the center of the worm body. The wavefront decomposition presented in Fig. 10e shows a significant amplitude drop of the dominant Zernike mode, 7 (coma). From the effective OTF of 3D-SIM, we can see more high frequency components after the sample aberration correction, which indicates a sharper image with more fine detail. 3D visualizations[47] of the axon before and after sample aberration correction can be seen in the supplementary movies. In the movie with system correction, fluorescence can be seen above the neuron from the cell body. This results from the lower SNR and degraded axial resolution without sample correction and cannot

be seen in the video with sample correction because the cell body is above the imaged volume.

Although direct wavefront sensing possesses high speed and good accuracy, it fails when the aberrations destroy the image of the focal spot array. For example, the beads under the worm body in Fig. 4 can only be corrected effectively using sensorless AO. But direct wavefront sensing fails because some spots were too strongly distorted or shifted to be detected in the Shack–Hartmann image (see Supplementary Fig. S6). This occurs when refraction is strong and the amplitude distribution in the back pupil plane is altered.

To evaluate the overall performance of AO-3DSIM, image SNR is calculated. The noise is defined as the STD around the maximum, which is defined as the signal. Overall, we see an improvement in the SNR for 3D-SIM of greater than 50%. The detailed results can be seen in Supplementary Table 2.

## Discussion
As SR microscopy becomes more mature, SR techniques are becoming fairly routine for imaging single cells in tissue culture. Extending these approaches to thick multicellular samples is a natural next step and will benefit many different areas of biological research from neuroscience, where synapses can be imaged

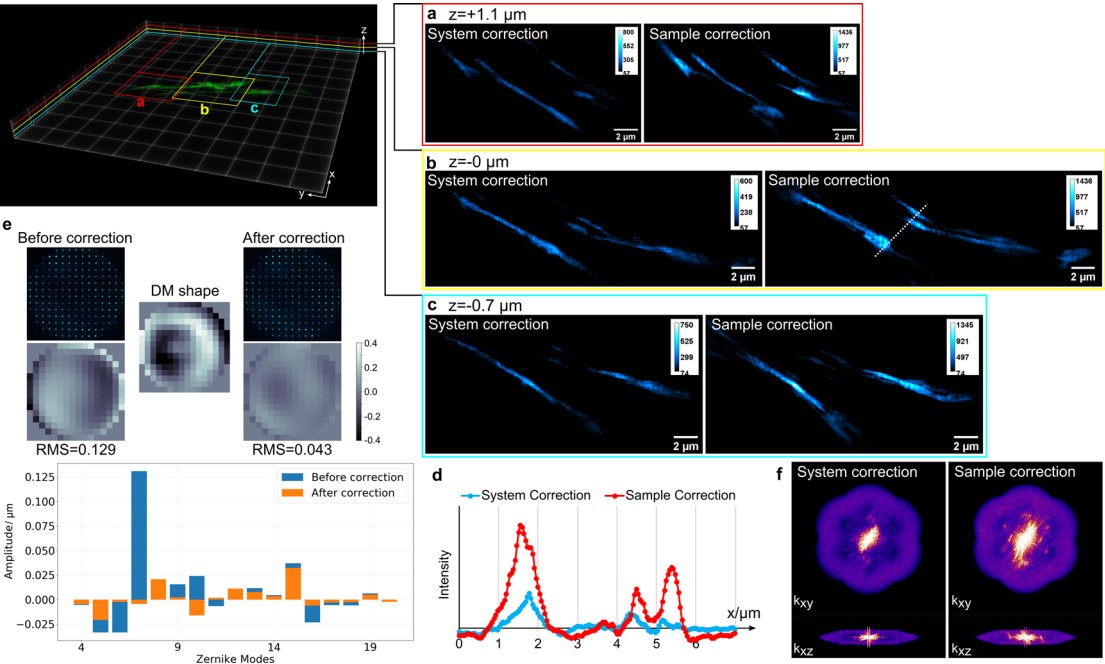

**Fig. 10 Live *C. elegans* with TBH-1::GFP expressed in RIC interneurons in the lateral ganglion.** The 3D-SIM images of the neural fiber before and after sample aberration correction at different z positions: **a** z = +1.1 μm. **b** z = 0.0 μm. **c** z = −0.7 μm. **d** The intensity profiles plotted along the white dotted line in 3D-SIM images. **e** The measured wavefronts and their Zernike mode decompositions before and after sample aberration correction and the corrective wavefront applied on the DM (units in microns). **f** Effective OTF of 3D-SIM images before and after sample aberration correction.

in detail only with resolution beyond the diffraction limit, to cancer research, where studying organoids will benefit from SR imaging of mitochondria and other cellular machinery. Unfortunately, aberrations can severely derail imaging in thick samples and lead not only to poor SNR, but also to incorrect imaging of morphology as can be seen, for example, in Fig. 9.

Here we have presented the combination of SIM with AO. While the improvement with AO is strongly sample dependent, the results can be substantial. We have demonstrated a lateral resolution of 150 nm and an axial resolution of 570 nm at a 670 nm emission wavelength when imaging through the body of an adult *C. elegans* at a depth of ~80 μm. The improvement in axial resolution is at least 50% and the improvement in SNR is more than 70%. Not only do the resolution and SNR improve, but, more importantly, the image fidelity improves as well. Under severe conditions, such as blurry images of AJs in *C. elegans*, confocal illumination can be used to assist the image quality assessment, removing the dependence on the object in the metric function. We have also implemented 3D-SIM with direct wavefront sensing with confocal illumination. We used the confocal spot as the "guide star." We demonstrated that the widefield image based sensorless AO method, the confocal spot based sensorless AO method, and confocal illumination based direct wavefront sensing AO method can all help to improve the image quality and fidelity of the 3D-SIM images. Each approach has strengths and weaknesses, and they can further be applied in conjunction.

AO has been applied to SIM[21,22], STED[48,49], and SMLM[26,50]. Previous combinations of SIM with AO have been limited to 2D[20] or have not been applied to full 3D-SIM imaging[36], only providing a resolution enhancement in the lateral plane. From our work, we see that the improvement to axial resolution in 3D-SIM is substantial. The axial resolution degrades more quickly with aberrations. This can be seen by examining the effect of aberrations such as coma and spherical aberration on the PSF; these

modes do not change the lateral size of the PSF central lobe although they may add side lobes and reduce the main lobe intensity, but they clearly increase the axial extent of the PSF. Some applications of AO in fluorescence microscopy have further relied on two-photon excitation to create the guide star for wavefront sensing[21,22]. While this has been proven to be an effective approach, it substantially adds to the cost and complexity. Here we have demonstrated a system with confocal illumination that only requires an additional beam path for excitation. A potential improvement to our system would be the addition of a galvo-system for scanning and descanning the confocal spot.

The time required for AO correction is 6 s, and the time to acquire the 3D-SIM raw data is ~90 s. Although these times can be reduced with further optimization of the microscope control software (Supplementary Table 2 contains the details of each acquisition.) Therefore, the time for AO correction is relatively modest compared to the time to acquire the SIM data. Furthermore, when correcting time-lapse data, the AO correction must not necessarily be performed before every image stack. Typically, the wavefront errors are caused by structures that are relatively static even during live imaging. The acquisition time can also be decreased for live imaging through the use of shorter exposure times, and, as has previously been noted, the time resolution is not limited by the time to acquire one stack but rather by the time to acquire one depth-of-focus[14].

AO-3DSIM is a promising method to acquire three-dimensional SR images in live thick samples with improved fidelity. Many different kinds of biological samples will benefit from the improved resolution in live imaging including imaging of *C. elegans* and other small model organisms, plant roots, plant–fungal interactions, and tissue spheroids which are important for three-dimensional cell culture. AO-3DSIM has enormous potential to help understand subcellular dynamics in organisms and tissues in vivo.

## Methods

**Three-dimensional structured illumination**. To separate the different copies (orders) of sample data and reconstruct the SR image, we take 15 raw images for each plane, which consist of three pattern orientations (0°, 120°, 240°), and five equally distributed phases $(0, \frac{2}{5}\pi, \frac{4}{5}\pi, \frac{6}{5}\pi, \frac{8}{5}\pi)$ for each angle. The three-dimensional data is acquired as a sequence of two-dimensional images as the sample is stepped through the focal plane. The copies of sample data are separated through a linear combination of acquired images, and the final three-dimensional image is obtained through a linear recombination of all post-processed and frequency-shifted images $\widetilde{D}_{j,m}(\mathbf{k})$, using a generalized Wiener filter:

$$\widetilde{S}(\mathbf{k}) = \frac{\sum_{j,m} a_{j,m} \left\{ \widetilde{H}_m\left(\mathbf{k}+m\mathbf{p}_j\right) \cdot \left[1 - G\left(\mathbf{k}+m\mathbf{p}_j\right)\right] \right\}^* \cdot \left[\widetilde{D}_{j,m}\left(\mathbf{k}+m\mathbf{p}_j\right) \otimes \widetilde{T}(\sigma)\right]}{\sum_{j,m} \left|\widetilde{H}_m^*\left(\mathbf{k}+m\mathbf{p}_j\right) \cdot \left[1 - G\left(\mathbf{k}+m\mathbf{p}_j\right)\right]\right|^2 + w^2} \cdot A(\mathbf{k})$$

(1)

where $w^2$ is the Wiener parameter, $m\mathbf{p}_j$ denotes $m$th order frequency component at the $j$th pattern orientation. $\widetilde{D}_{j,m}(\mathbf{k})$ is the FT of the data for the mth order and jth angle. $\widetilde{H}_m(\mathbf{k})$ is the OTF for the $m$th order; the OTF for the first order consists of two copies of the zeroth order OTF shifted in the axial direction. $a_{j,m}$ is an experimentally determined complex weight for each component. $G(\mathbf{k}+m\mathbf{p}_j)$ is a top-hat notch filter with a diameter of nine pixels, which suppresses the overemphasized zero frequency of each OTF copy, removing the residual stripe artifacts and enhancing the optical sectioning[51,52]. $T(\gamma)$ is a Tukey window function ($\gamma = 0.08$) that is used to eliminate the edge effect in the fast Fourier transform (FFT) due to edge discontinuities. $A(k)$ is the apodization function to cancel the ringing of high frequency spectral components caused by the Wiener filter. We used a modified three-dimensional triangle function $A(k) = [(1 - a_{xy} \cdot k_{xy}/k_{cutoff\_xy}) \cdot (1 - a_z \cdot k_z/k_{cutoff\_z})]^n$, where the cutoff frequencies $k_{cutoff\_xy} = 4 \cdot NA/\lambda$, $k_{cutoff\_z} = NA^2/\lambda$. The parameters $a_{xy} \in [0.5, 1]$, $a_z \in [0.5, 1]$, and $n \in [0.5, 2]$ are tuned empirically to balance the resolution loss and the artifact elimination. All functions are defined in a three-dimensional volume with the size of $1024 \times 1024 \times n_z$, in which $n_z$ is determined by the number of slices in the z-stack.

**Sensorless AO**. For sensorless AO, the sample is illuminated either with widefield illumination by turning all SLM pixels on or with the confocal illumination path. For the sensorless AO method, three/five images are taken for the trial Zernike mode, $Z_l$, at three/five different strengths $(-a, 0, a)/(-2a, -a, 0, a, 2a)$. For each image, the metric function value, $M_l(x)$, is calculated. A quadratic fit is applied to $x = (-a, 0, a)/(-2a, -a, 0, a, 2a)$ and $y = M_l(x)$. We typically initiated our experiments with the step size $a = 0.2\,\mu m$. While, for large aberration cases, such as the image of beads under a worm body in Fig. 2, we enlarged the step size to $a = 0.4\,\mu m$. The peak of the parabolic fit is set as the optimal correction strength for the corresponding Zernike mode and the mirror shape is adjusted accordingly. This procedure is repeated on all modes to be corrected to get the optimal strength for each mode separately. Then all modes are combined linearly to form the final corrected wavefront. If needed, the above process can be repeated a set number of times with smaller step size or until the image no longer improves. The photobleaching effect during the sensorless AO iterations is examined (see Supplementary Fig. S7). The results show minimal photobleaching measured over 130 image frames for GFP or over 65 frames for organic dye. In most experiments, our image-based corrections are focused on the Zernike modes 4, 5, 6, 7, and 10 that are responsible for astigmatism, coma, and spherical aberration, and we need to take 25 image frames for one AO iteration, which costs ~6 s including the computation, DM configuration, and 100-ms exposure time for each image. The time cost can be reduced to ~3.5 s by only taking three images for each mode, which meets the minimum number of data points required for a quadratic fit. And in certain cases, such as imaging *C. elegans* as shown in Fig. 6, we can make the iteration even faster by setting the exposure time to 50 ms for each image, which reduced the time cost to less than 3 s for one iteration of five modes.

The core of the sensorless AO method is the image quality metric. As we can see from the effective OTFs in Fig. 1a, although different aberrations affect the PSF and OTF differently, the strength of the OTF always gets reduced more at higher frequencies. Therefore, we use a spatial frequency based metric function, $M$, defined as[26]:

$$M = \frac{\sum_{u,v} \left|\widetilde{I}(u,v)\right|(1 - g_h(u,v)) \mathrm{circ}\left(\frac{\lambda}{2 \cdot NA}\sqrt{u^2 + v^2}\right)}{\sum_{u,v} \left|\widetilde{I}(u,v)\right| g_l(u,v) \mathrm{circ}\left(\frac{\lambda}{2 \cdot NA}\sqrt{u^2 + v^2}\right)}$$

(2)

where $g(u,v) = e^{-\frac{u^2+v^2}{2\sigma^2}}$ is the Gaussian function, and $\widetilde{I}(u,v)$ is the frequency spectrum of the widefield image.

The two Gaussian filters $g_h$ and $g_l$ can be tuned according to the frequency spectrum of the image to adjust the weight of the high and low frequency components in the numerator and denominator. For the optimal sensitivity and accuracy, in most cases, we choose a slightly larger FWHM for the high frequency filter in the numerator to emphasize high frequency components higher than $NA/2\lambda$, and a smaller FWHM for the low frequency filter so that frequencies near

0 are emphasized in the denominator. Moreover, the assessment accuracy of the metric function also relies on the image spectrum distribution and the SNR. An ideal object for evaluating the metric is a sub-resolution fluorescent bead or other bright point-like object because these contain all spatial frequencies uniformly. However, in biological samples, such objects are rarely present. If the frequency spectrum of the widefield image is highly concentrated within a limited range, or contains strong high frequency noise, the metric function will lose its efficiency, or even give erroneous results. To enhance the robustness, a confocal spot close to diffraction limit can be used for illumination to remove the object dependence and improve the efficiency of the sensorless AO method, as shown in Fig. 7. And because the image of the confocal spot is close to the PSF, we can even use a simple metric function such as the peak value instead of a frequency based metric function without losing any accuracy.

Practically, complete correction over a large field of view is rarely achievable, especially on thick biological samples. When the existing aberrations contain anisoplanatic terms, the AO system will fail due to an averaged aberration estimation over the whole widefield image, which could be far from ideal. Therefore, for better performance, a region of interest (ROI) in the image is chosen as the target. The correction procedure is then conducted on this target area. This has proven to be effective as shown in Fig. 5, with the disadvantage that the image quality is degraded outside the ROI.

**Direct wavefront sensing AO method**. We measure the wavefront with a SHWFS and correct the wavefront in a closed loop system as is commonly done in astronomy[53] and has been applied in microscopy[22,30,54,55]. The aberration-free image is set by applying the sensorless AO method on 200-nm fluorescent beads on a coverslip. We then record the reference wavefront corresponding to an aberration-free image on the SHWFS. The biological sample is imaged, measuring the wavefront on the SHWFS using the confocal spot as the guide star. The correction could require several iterations for the wavefront to converge to the reference. In this method, the microscope system is switched to confocal illumination for wavefront correction, and then switched to SI illumination for imaging.

From the raw Shack–Hartmann images, we determine the wavefront gradients using cross-correlations with the reference Shack–Hartmann image. We use the FT based wavefront reconstruction method[56] to reconstruct the aberrated wavefront. The aberrated wavefront is then decomposed into Zernike modes, and tip, tilt, and defocus are removed to generate the correction wavefront. A large portion of the tip/tilt is due to the shift between the reference SH image and the measured SH image, and the defocus term is affected by light out of the focal plane due to the large lenslet depth of focus. To decompose the wavefront, pixelated Zernike modes with the same size as the measured wavefronts are generated from the Zernike mode definition and then orthogonalized using a Gram–Schmidt procedure to removed residual crosstalk due to the digitization.

**SNR calculation**. To calculate the SNR, we first crop an ROI around an identifiable object (bead, nerve fiber, etc.) surrounded by a relatively dark background. We calculate the mean value of the pixels forming the object and divide by the STD of the surrounding area, which is defined as the pixels with values lower than a threshold determined by the adaptive threshold algorithm in the python sci-kit image package (threshold_local() in skimage.filters, https://scikit-image.org/ docs/0.18.x/api/skimage.filters.html#skimage.filters.threshold_local).

**Experimental setup**. The AO-3DSIM system is built on an Olympus IX71 inverted microscope with a Prior Proscan XY Stage and a Prior 200-nm travel NanoScan Z stage for sample movement and focusing.

Excitation light (Cyan 488 nm—Newport, or 647 nm—Coherent OBIS Lasers) is reflected by a polarizing beam splitter where the s-polarized light (perpendicular to the optical table) is collimated and expanded 5x by a telescope lens pair (L1, $f_1 = 40$ mm, and L2, $f_2 = 200$ mm), achieving a quasi-uniform light intensity around the beam center. An iris with a diameter of ~20 mm is inserted after L2 to limit the beam size. The linear polarized beam is sent through a pattern generating unit consisting of a $2048 \times 1536$ pixel spatial light modulator with pixel size of 8.2 μm (Forth Dimension QXGA-3DM), a polarizing beam splitter cube (10FC16PB.3, Newport), and an achromatic half-wave plate (AHWP10M-600, Thorlabs). In this work, we use a biased 15-pixel binary pattern (6 pixels ON, 9 pixels OFF) and a lens L5 ($f_5 = 300$ mm) to generate an illumination pattern with period of 341.67 nm at the objective focal plane, which provides the resolution enhancement of $1 + [\lambda/(2\cdot NA)]/341.67 = 1.818$ for red fluorescence (670 nm) in principle. For the green fluorescence (515 nm), we use a biased 10-pixel binary pattern (4 pixels ON, 6 pixels OFF) and a lens L5 ($f_5 = 250$ mm). The period at the objective focal plane is then 273.33 nm, corresponding to a resolution enhancement of $1 + [\lambda/(2\cdot NA)]/273.33 = 1.78$. To maximize the modulation depth of the sinusoidal pattern, the polarization state of the illumination light must be normal to the direction of pattern wave vector for all three angles. This is controlled by a half-wave plate mounted on a fast-motorized polarizer rotator (8MRU, Altechna). The dichroic beamsplitter (Semrock BrightLine® FF410/504/582/669-Di01) used to reflect the illumination beam is designed with low polarization dependence so that the orthogonal relationship between polarization state and pattern wave vector is

maintained. A mask on an intermediate pupil plane is used to block all unwanted diffraction orders except for the 0 and ±1 order. For strengthening the lateral frequency component in the pattern, the diffraction grating is intentionally designed so that the intensity of the ±1 order is ~3x higher than that of the 0 order. The three beams interfere at the focal plane forming a structured illumination pattern in three dimensions.

The fluorescent emission light from the specimen is collected by the objective (Olympus UPlanSApo 60x water immersion objective with NA of 1.2) and exits the microscope body from the left-side port. A 250-mm lens projects the objective pupil plane onto the DM (ALPAO DM69, 69 actuators and 10 mm diameter), which exactly matches the 1.2NA back pupil size of the system. The DM is conjugate to the back pupil plane to provide correction over the whole field of view. The corrected fluorescent image is then directed to the Andor EMCCD camera (DV887DCS-BV with 14 bit ADC) through the dichroic beamsplitter. The total magnification of the system is 180x, so the effective pixel size of the final image is 89 nm. An emission filter (Semrock BrightLine® quad-band bandpass filter, FF01-446/523/600/677-25) and a notch filter (Semrock StopLine® quad-notch filter, NF03-405/488/561/635E-25) are put before the camera to block unwanted light, assuring a low background and noise level.

The direct wavefront sensing system consists of a home-built SHWFS and a confocal illumination scheme. The laser beam after the polarized beamsplitter is focused by a lens (L3, $f_3 = 17$ mm), spatially filtered by the first pinhole (PH1) with a diameter of 5 μm, and collimated by a lens (L4, $f_4 = 300$ mm). The rotatable mirror $a$ is switched to its position 1 to pass the collimated beam to the illumination path. After the lens pair (L6, $f_6 = 100.7$ mm, and L7, $f_7 = 250$ mm), the collimated beam reaches the DM with a beam diameter larger than 10 mm, ensuring a full coverage of the back pupil plane. A second pinhole (PH2) is put at the first image plane after the tube lens, acting as a confocal pinhole to block the out-of-focus light. To ensure a complete transmission of the aberrated PSF, which is laterally enlarged by the wavefront distortion, we choose a pinhole with a diameter of 500 μm for the second pinhole, PH2. Details about the pinhole in confocal SHWFS can be found in Supplementary Fig. 8. The emission path needs to be perfectly aligned with the confocal illumination path to pass through the second pinhole PH2, otherwise either the illumination or emission beam will be blocked by the pinhole. The emission light beam is directed to the direct wavefront sensing path by the rotatable mirror $b$ at its position 2. The reflected emission light is sent through a lens (L10, $f_{10} = 170$ mm) to project the conjugated pupil plane onto the lenslet array (Thorlabs, MLA150-5C). The conjugated pupil size at the lenslet array will be 2.27 mm, which covers an area of ~14 lenslets in diameter. A 1:1 doublet pair (Thorlabs - MAP105050-A) reimages the focal plane of the lenslet array onto the WFS camera (PCO.Edge 4.2LT Monochrome sCMOS camera), forming the image of the focal spot array. From geometrical considerations, the maximum phase change that can be measured by this SHWFS is $\pm d^2/2f_{ML} = \pm 2.05\,\mu m$, where $d = 146\,\mu m$ is the lenslet diameter and $f_{ML} = 5.2$ mm is the lenslet focal length.

To measure the DM actuator influence functions and monitor the wavefront applied to the DM, we added a separate laser beam (520 nm—PL203 Compact Laser Module, Thorlabs) to mimic the emission beam with a flat wavefront. The collimated laser beam enters the optical path after the lens L8, by switching the rotatable mirror $c$ to its position 2. The laser beam follows the same path as the emission light beam to the SHWFS.

**Sample Preparation.** We have complied with all relevant ethical guidelines for animal use and research.

1. Beads under *C. elegans*[57]
   The diluted 100-nm fluorescent beads (T7279, Life Technologies, Tetra-Speck™ Microspheres, 0.1 μm, fluorescent blue/green/orange/dark red) were dried onto charged slides (9951APLUS-006, Aqua ColorFrost Plus). Wild-type *C. elegans* were placed above the beads with 10 μL of a 50 mM solution of phosphatase inhibitor Tetramisole hydrochloride (T1512, Sigma-Aldrich) to inhibit worm movement. Before the Tetramisole solution dries completely, 10 μL of glycerol was put on the slide, and the coverslip was mounted and fixed carefully.

2. Cell culture.
   α-TN4 lens epithelial cells were grown on poly-L-lysine coated coverslips, fixed with 4% paraformaldehyde in phosphate-buffered saline (PBS) pH ~7.4. And then the fixed cells were incubated in 1 mL Phalloidin-iFluor 647 (ab176759) solution (1 μL 1000x stock solution in 1 mL PBS) under 4 °C overnight to stain the F-actin. After staining, the cells were washed with PBS three times, and mounted with antifade medium (VECTASHIELD® H-1000) for imaging.

3. *C. elegans*.
   The *C. elegans* strains MT9971 (*nIs107* [*tbh-1*::GFP + *lin-15(+)*] III) and BR2958 (*ceh-16(lg16)*; *jcIs1* [*ajm-1*::GFP + *unc-29(+)* + *rol-6(su1006)*] IV; *ngEx1*[*ceh-16*::GFP]) were obtained from the Caenorhabditis Genetics Center. Worms were grown on 3x nematode growth media (NGM) agar plates (3 g/L NaCl, 7.5 g/L bacto peptone, 20 g/L agar, 1 mM MgSO4, 1 mM CaCl2, 5 mg cholesterol in ethanol, and 25 mM K2HPO4) and fed OP50 bacteria that was grown on the agar plates.

To mount worms: a drop of 2% agarose was placed onto a clean slide, and then covered with another clean slide to flatten the agarose. After the agarose becomes solid, the top slide is then gently shifted to separate the two slides, allowing the agarose pad to adhere to one of the two slides. Ten microliters of 20 mM sodium Azide (NaN3) and 10 μL of 50 mM Tetramisole solution, both dissolved in M9 buffer (3.0 g/L KH2PO4, 6.0 g/L Na2HPO4, 0.5 g/L NaCl, 1.0 g/L NH4Cl), were placed on the center of the agarose pad. The worm to be observed was then transfer into the drop with a worm pick. A clean coverslip was gently put on and sealed with nail polish.

4. *M. oryzae* growing inside rice cells.
   *M. oryzae* transgenic strain CKF4019, expressing GFP retained in the ER lumen, was generated by transforming *M. oryzae* wild-type strain O-137 with plasmid pCK1724[40]. The plasmid carries GFP fused at 5′end with the sequence encoding the BAS4 signal peptide and at 3′ end with the sequence encoding the ER retention signal peptide HDEL. A rice sheath inoculation method was used to prepare optically clear rice tissue with the fungus[40]. The rice sheaths from 20-day old rice cultivar YT16 were excised to about 8 cm in length, and subsequently the inner epidermal tissue of the excised sheath was inoculated with CKF4019 conidial suspension ($1 \times 10^5$ conidia/ml in water). At 30 h post inoculation, the inoculated rice sheath was hand-trimmed with a razor blade to produce optically clear leaf sheath slices with 3–4 cell layers deep and about 60 μm thick. The trimmed sheath was mounted on a slide in water under a coverslip for live-cell imaging.

**Statistics and reproducibility.** For all the structures imaged in the manuscript (alpha-TN4 actin, C. elegans RIC interneuron, C. elegans AJs, and M. oryzae ER) at least three samples were imaged to confirm that the images improved with the application of SIM and AO. For the measurements of bead images in Fig. 4, five beads were measured so that we saw a statistical difference comparing most cases (with and without structured illumination and with and without AO).

**Reporting Summary.** Further information on research design is available in the Nature Research Reporting Summary linked to this article.

## Data availability

Source data are provided with this paper. The image data (raw data, reconstructed images, and image metadata) are available on https://www.ebi.ac.uk/biostudies/studies/S-BSST629.

## Code availability

The 3D-SIM Reconstruction Software[58] (https://doi.org/10.5281/zenodo.4690773) and the Microscope control software[59] (https://doi.org/10.5281/zenodo.4690769) are available at https://github.com/Knerlab.

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

## Acknowledgements

We thank James D. Lauderdale for help with tissue culture. The graphic rendition of the neurons and cells in *C. elegans* (Figs. 6) are by courtesy of the WormAtlas[60]. This research was supported by the National Science Foundation under grant DBI-1350654 (P.K.) and the Basic Research to Enable Agricultural Development (BREAD) program (Award number 1543901) from the National Science Foundation (C.H.K.), and NIGMS grant R01GM134359 (E.T.K.). Some *C. elegans* strains were provided by the CGC, which is funded by NIH Office of Research Infrastructure Programs (P40 OD010440).

## Author contributions

P.K devised and supervised the project. R.L. and P.K. built the microscope and wrote the software. R.L. prepared the tissue culture samples, acquired and analyzed all the data. E.T.K. provided expertize and assistance with *C. elegans* biology and imaging and provided some of the *C. elegans* strains. J.Z. and C.H.K. prepared the *M. oryzae* samples and provided expertize and assistance with *M. oryzae* biology and imaging. R.L. and P.K. wrote the manuscript with assistance from all authors.

## Competing interests

The authors declare no competing interests.

**Additional information**

