## [Peer Review File · Nature Communications]

Reviewers' Comments:

Reviewer #1:

Remarks to the Author:

In this ms, Kner and coworkers demonstrate a combination of sensorless adaptive optics (AO) and 3D structured illumination microscopy (SIM), and their combined method to several different biological samples. 3D SIM allows resolution enhancement over the diffraction limit in principle, yet in many samples, aberrations due to the sample (or instrument) prevent super-resolution imaging (or even diffraction-limited imaging) in practice. The authors convincingly show, in the presented samples, that improvement in imaging and SNR is possible with AO. 3D SIM as they have implemented it is not novel, and the general class of AO that they demonstrate (sensorless AO) has been demonstrated before too. Also, the particular image metric they use is likely to be useful mainly for 3D SIM and not necessarily other super-resolution techniques. Nevertheless, the combination of the methods they represent is more powerful than either of the component methods alone, and the combination is likely to be of interest to the SIM field.

The biological demonstrations are somewhat contrived (beads under anesthetized *C. elegans* are useful for a technical demonstration only) or weak (the nerve fiber images are improved but the improvement doesn't seem to offer much of a qualitative improvement to the interpretation of the image) – in particular it would have been nice to see some time-lapse experiments or other examples in which the AO made a substantive difference to the biology under investigation.

The authors also assert in the Introduction that their sensorless method allows AO correction with 'minimal photobleaching' – yet this is never demonstrated in their paper, and indeed for the data shown in Fig. 5, the authors point out that the GFP fluorescence cannot survive too many AO correction loops. The authors should either demonstrate that their method indeed enables AO correction with minimal photobleaching or remove this assertion from the paper.

Other comments:

The authors refer to 'Full 3D imaging' in the title and 'full 3D SIM' in the abstract - and elsewhere in the text. What do they mean by 'full'? I would suggest omitting this word in the interest of clarity.

'*C. elegans* worm' is redundant

It is a bit puzzling that the XZ image sections in Fig. 1d look so isotropic, given the known asymmetry of the axial point spread function. Is the apparent isotropy due to the max projection that is shown in the figure? This might be worth clarifying in the figure caption.

Why is there an effective 'doubling' of the beads in z when spherical aberration is present, i.e. lowest row of Fig. 1d, xz projection?

Does the overlap amplitude show up in the equation for the Fourier transform of the 3D image, $S(k)$? It would be useful to define precisely what 'overlap amplitude' is (i.e. mathematically), given that it is plotted in Fig. 1f.

How many images, on average, are required? E.g. if 5 images are taken for 5 Zernike modes, does this mean on average 25 images are required for the data presented in the paper? I would like to see a table with this information for every dataset presented in the paper. Is the wavefront estimated 1x per volume or for every plane in the volume? Presumably the latter would offer a more accurate AO correction, especially for thick samples, at the expense of increased photobleaching/damage. Please clarify.

A related point – the timing data presented in Table 2 are actually somewhat misleading – the true 'exposure time' is in fact likely much longer than what is presented in this table, which I'm

guessing is the time to acquire an image after the appropriate correction wavefront is applied to the DM. Please clarify this point in the caption and description of that table.

The SNR improvement claimed in Table 2 is likely an overestimate if the 'signal' is defined as the maximum pixel value in a region (which could be due to noise – a better metric might be the background-subtracted average in some region divided by the standard deviation in the same region). Also, please better define how the std is calculated – 'around the maximum' is pretty vague.

What exactly do the authors mean when they claim that imaging neuron filaments in *C. elegans* is 'severe'?

A clear description of the drawbacks of this method (presumably a small field of view, photobleaching due to the relatively large # of images required, slow correction due to the large number of images and thus worsened performance in moving samples, etc) would add value to the discussion. Even better if the strengths/weaknesses of the method are compared against the direct wavefront sensing 2P method as this method is generally considered state of the art.

Would be useful to describe in more detail what the arrows represent in Fig. 2a, in the caption – this appears to be the polarization vectors applied in illumination (and how they are generated) but the casual reader may not appreciate this.

Is there a reason that the maximum illumination spatial frequency was not used for the SIM, as in previous SR-SIM papers? Presumably this would result in a greater resolution improvement, in principle.

How much greater is the intensity (or illumination power) for the +/-1 orders vs. the zero order beam?

The fits in Fig. 5e don't fit the experimental points particularly well. This makes me wonder if the parabolic assumption is in fact an oversimplification, and if better results might be obtained with a more complex model/fit. Please clarify.

In Fig. 6d, the image looks overall weaker in intensity than Fig. 6c, is this due to photobleaching? In fact it looks like features in Fig. 6c are actually missing in Fig. 6d. Thus, I am not convinced that correcting the sample aberration offers a significant advantage over the static system correction. I also am not convinced that the image in Fig. 6c is significantly fuzzier than in Fig 6d – in fact it looks sharper (which might be due to noise). It's also not clear that the image 'looking smoother' is an indication that Fig. 6d is 'better' than Fig. 6c as the authors imply, i.e. if higher spatial frequency components are restored it might in fact be that the fully corrected image looks less smooth.

Reviewer #2:

Remarks to the Author:

Lin et al present adaptive optics (AO) applied to 3D structured illumination microscopy, and its application to various samples. They show that AO can be of great benefit to SIM reconstructions, both in suppressing artifacts, but also in obtaining high spatial resolution. The AO uses iterative optimization of Zernike modes using an image metric. In addition, the illumination can be patterned to add "structure" to the sample to help the algorithm converge, which is inspired by the IsoSense paper (Žurauskas, M., et al., *Optica*, 2019. 6(3): p. 370-379.).

In contrast to other publications that have combined SIM and AO, the authors claim that this is the first application of 3D SIM (three beam interference and 3D data acquisition) and that it goes

further than the IsoSense publication in terms of the complexity of the aberrations.

To my understanding, the IsoSense microscope is also a 3D SIM type, but I agree with the authors that in that publication, no serious aberrations were corrected (it looked more like correction of systems aberrations). In that sense, the authors' publication shows indeed more of the potential of AO and 3D SIM and the detailed results can serve as a reference what kind of gains can be expected. As such I found the manuscript interesting, albeit a bit incremental on previous publication.

larger issues:

The authors claim that they can correct large aberrations. The largest aberrations occur for the c-elegans, however they are of low complexity (it is mainly dominated by astigmatism). The magnitude is still lower than one wavelength, so it can also be debated if this is large. I am not fully convinced that the iterative scheme can deal well with large and complex aberrations, as the starting image will be of low SNR and contrast, and the sensitivity to the aberration probing will become low. For large aberrations, a Shack Hartmann sensor might be more robust, as it does not suffer from decaying image contrast (or other image metric). I understand that a Shack Hartmann comes with its own limitations, but I am not convinced that the iterative sensorless scheme is superior. To my knowledge, Zernike mode based sensorless AO has only been shown for relatively benign aberrations that were neither complex nor too far from the aberration free case. So I invite the authors to reconsider the phrasing of the magnitude and complexity of aberration correction.

Another concern is that the authors only do aberration corrections for relatively basic, low order aberrations (coma, astigmatism, spherical). This essentially prevents them from correcting any more complex, higher order aberrations. In other AO work, much higher aberration modes have been considered (Wang, K., et al. Nature Communications, 2015.). This leaves the concern that the authors picked some examples where this low complexity wavefront correction works well (i.e. correction of Astigmatism on the c-elegans).

Would it be possible to make an experiment with more complex aberrations? Maybe using a phantom to introduce higher order aberrations? Otherwise some claims need to be revised.

Another interesting point is that the aberration loop runs separately for each aberration mode and then at the end the ideal coefficients are computed. Would it make more sense to update each mode once it has been tested, such that subsequent modes start with less overall aberrations (and hence higher SNR and contrast)? My intuition is that with such a procedure, and the addition to run the whole optimization multiple times, sensorless AO could pull itself out of a bit more complicated aberrations.

For the Neuronal imaging, the authors applied a dot like pattern for illumination. Does the mask for 3D SIM need to be removed for that or how is the pattern generated? The authors should also show a Fourier transform with and without the dot pattern applied to better illustrate how it can increase the frequency content. As a further question, do the authors assume that the dot pattern is not distorted by the aberrations? Can that be further justified?

Smaller points:

There are different definitions for numbering Zernike modes. I assume the authors use the Noll index. This has to be more clearly defined. Likewise, the authors are a bit imprecise about their AO algorithm. What is the step size $(-2aa, -aa, 0, aa, 2aa)$? Is this in microns? Please give this number also in radians, to give a better idea how this scales to the wavelength.

It is not clear what the values on the wavefront sent to the mirror are (e.g. Figure 3i, 4f 5f 6h, microns? Radians?). Scaling it to 2π radians might be most instructive.

Table 1: any explanation why the gain in axial resolution in SIM with AO is so much more dramatic in the z-direction? In widefield with and without AO, the gain is rather small.

The authors compute signal to noise, although it is not clear to me how this is done. The text reads: "The noise is defined as the standard deviation (STD) around the maximum, which is defined as the signal. The image SNR." Is this performed in real or reciprocal space? The only related method I know is measuring standard deviation in frequency space outside of the OTF and comparing it to the DC frequency component. The authors have to better explain their method.

Overall, enthusiasm for this manuscript is high, but I recommend strengthening and clarifying the manuscript.

January 18, 2021

Manuscript NCOMMS-19-1472556-T, “Full three-dimensional imaging deep through multicellular samples with subcellular resolution by structured illumination and adaptive optics” by Lin et al.

Response to reviewers. General comments

We thank the reviewers for their comments and note that, while they had substantive criticisms of the manuscript, both reviewers considered the paper interesting and significant. Reviewer 2 noted that “enthusiasm was high”.

In order to address the reviewers’ concerns we have changed our experimental system. We have modified our microscope so that it is now capable of widefield, confocal, and structured illumination, and we have added direct wavefront sensing capability. We now present results with three different approaches for adaptive optics correction: sensorless AO with widefield illumination, sensorless AO with confocal illumination, and direct wavefront sensing (using a Shack-Hartmann Wavefront Sensor) with confocal illumination. With these methods we show AO correction on a variety of samples and demonstrate, we believe, that AO can have a significant effect on the image quality, allowing more accurate interpretation of the images.

We therefore have made substantial changes to the manuscript. In the methods section, we discuss the different methods for wavefront correction. We have modified figure 1 to respond to a question from the reviewers. We have modified figure 2 to illustrate the new beam paths in the modified microscope. Figures 5 and 6 have been modified to respond to comments from the reviewers. Figures 7, 8, 9, and 10 are new figures demonstrating the new correction approaches. We have modified the conclusion to provide more discussion of the correction ability. Finally, we have moved the methods section after the results to conform better to the style of Nature Communications.

Below we respond to specific comments from the reviewers. Their comments are in black text and our responses are in blue text.

Reviewer #1 (Remarks to the Author):

In this ms, Kner and coworkers demonstrate a combination of sensorless adaptive optics (AO) and 3D structured illumination microscopy (SIM), and their combined method to several different biological samples. 3D SIM allows resolution enhancement over the diffraction limit in principle, yet in many samples, aberrations due to the sample (or instrument) prevent super-resolution imaging (or even diffraction-limited imaging) in practice. The authors convincingly show, in the presented samples, that improvement in imaging and SNR is possible with AO. 3D SIM as they have implemented it is not novel, and the general class of AO that they demonstrate (sensorless AO) has been demonstrated before too. Also, the particular image metric they use is likely to be useful mainly for 3D SIM and not necessarily other super-resolution techniques.

Nevertheless, the combination of the methods they represent is more powerful than either of the component methods alone, and the combination is likely to be of interest to the SIM field.

We thank the reviewer for his comments. We agree that the implementation of 3D SIM is not novel, although we have modified the reconstruction algorithm. We have now modified our setup to implement three different methods of AO correction: sensorless AO over an ROI, sensorless AO with confocal illumination, and AO correction with a SHWFS. While these different approaches have been tried before, we would argue that the combination is significant. Also we believe the results are important; we demonstrate that AO has a substantial impact on the resolution in the axial direction in SIM which has not to our knowledge previously been demonstrated. The metric we use is from a previous paper of ours and we have demonstrated that it is effective in AO-STORM.

The biological demonstrations are somewhat contrived (beads under anesthetized *C. elegans* are useful for a technical demonstration only) or weak (the nerve fiber images are improved but the improvement doesn't seem to offer much of a qualitative improvement to the interpretation of the image) – in particular it would have been nice to see some time-lapse experiments or other examples in which the AO made a substantive difference to the biology under investigation.

We image five different samples. Beads under a worm is admittedly a bit contrived but imaging beads is an important demonstration for methods where resolution is an important metric so we feel that this is justified. We believe it is quite common to show the result of imaging beads in papers on both AO and SIM. Imaging tissue culture cells (in our case an alpha-TN4 cell) is also common in microscopy AO papers. It is helpful to show the correction of system aberrations and the effect of aberration correction on a known biological sample.

The *C. elegans* imaging and the imaging of rice blast fungus are not contrived at all. These are samples that are important to our collaborators for ongoing biological research. We now show wavefront correction with both sensorless AO and direct wavefront sensing, and we see substantial changes to the images that, in our opinion, do make a substantive difference to interpreting the biology under investigation. As particular examples, Fig. 6 d & e, Fig. 7 a & b and Fig. 9 b (the zy views) all show differences to the morphology as well as signal intensity and noise.

The authors also assert in the Introduction that their sensorless method allows AO correction with 'minimal photobleaching' – yet this is never demonstrated in their paper, and indeed for the data shown in Fig. 5, the authors point out that the GFP fluorescence cannot survive too many AO correction loops. The authors should either demonstrate that their method indeed enables AO correction with minimal photobleaching or remove this assertion from the paper.

We have updated Table 2 to include data on the number of images required for both the SIM and the wavefront correction. We now include plots in the supplementary data (Fig. S6) showing the amount of photobleaching. The maximum number of images we use for AO correction is 50. We feel that Fig. S6 supports the claim that photobleaching is minimal during AO correction.

Other comments:

The authors refer to 'Full 3D imaging' in the title and 'full 3D SIM' in the abstract - and elsewhere in the text. What do they mean by 'full'? I would suggest omitting this word in the interest of clarity.

We used the word 'full' to emphasize that we demonstrate increased resolution in all three dimensions in contrast to previous work that we cite in the introduction. But we agree that the meaning is unclear and have removed the word 'full'.

'C. elegans worm' is redundant.

We have fixed this throughout the manuscript.

It is a bit puzzling that the XZ image sections in Fig. 1d look so isotropic, given the known asymmetry of the axial point spread function. Is the apparent isotropy due to the max projection that is shown in the figure? This might be worth clarifying in the figure caption.

The axial scaling was compressed so that the artifacts in the axial direction could be seen. We have redone the figure so that the axial scale is compressed by a factor of 2 relative to the lateral scale making the elongation of the PSF more apparent. We have also added scale bars to the figures. We have also changed the example images in this figure to include more objects, giving a better understanding of the effect of aberrations on complex objects.

Why is there an effective 'doubling' of the beads in z when spherical aberration is present, i.e. lowest row of Fig. 1d, xz projection?

The 'doubling' is due to the interaction of 3D SIM processing and spherical aberration. Spherical aberration elongates the PSF in the axial direction. When this effect is combined with the addition of multiple OTF copies in SIM processing, the 'doubling' that you see occurs. These distortions underline the importance of aberration correction in SIM. We have changed Fig.1 so, this doubling is not as apparent because the axial range has changed.

Does the overlap amplitude show up in the equation for the Fourier transform of the 3D image, $S(k)$? It would be useful to define precisely what 'overlap amplitude' is (i.e. mathematically), given that it is plotted in Fig. 1f.

Yes. This is a good point. We have corrected the equation for $S(k)$ to include the weighting, $a_{j,m}$, of each term. This weighting is determined by the overlap. We have changed the equation (now on page 16) and added the following sentence to the following paragraph:

$a_{j,m}$ is an experimentally determined complex weight for each component.

How many images, on average, are required? E.g. if 5 images are taken for 5 Zernike modes, does this mean on average 25 images are required for the data presented in the paper? I would like to see a table with this information for every dataset presented in the paper. Is the wavefront estimated 1x per volume or for every plane in the volume? Presumably the latter would offer a

more accurate AO correction, especially for thick samples, at the expense of increased photobleaching/damage. Please clarify.

A related point – the timing data presented in Table 2 are actually somewhat misleading – the true ‘exposure time’ is in fact likely much longer than what is presented in this table, which I’m guessing is the time to acquire an image after the appropriate correction wavefront is applied to the DM. Please clarify this point in the caption and description of that table.

We now include in Supplementary Table 2 all the information on how many images are taken for wavefront correction and for SIM. Along with the exposure time per raw frame, this provides all the information needed to calculate the various ‘exposure times’ associated with acquiring each image. Table 2 in the original manuscript has now been moved to the Supplementary Information and is Supplementary Table 2.

The SNR improvement claimed in Table 2 is likely an overestimate if the ‘signal’ is defined as the maximum pixel value in a region (which could be due to noise – a better metric might be the background-subtracted average in some region divided by the standard deviation in the same region). Also, please better define how the std is calculated – ‘around the maximum’ is pretty vague.

We thank the reviewer, and we have changed the SNR calculation and define more accurately how the SNR is calculated. We have now included a section in the methods describing the SNR calculation.

“SNR Calculation

To calculate the SNR, we first crop an ROI around an identifiable object (bead, nerve fiber, etc.) surrounded by a relatively dark background. We calculate the mean value of the pixels forming the object and divide by the STD of the surrounding area which is defined as the pixels with values lower than a threshold determined by the adaptive threshold algorithm in the python scikit-image package (`threshold_local()` in `skimage.filters`, https://scikit-image.org/docs/0.18.x/api/skimage.filters.html#skimage.filters.threshold_local.)”

What exactly do the authors mean when they claim that imaging neuron filaments in *C. elegans* is ‘severe’?

Yes, this term was unclear. We have replaced that sentence with the one below.

“Unfortunately, aberrations can severely derail imaging in thick samples and lead not only to poor SNR, but also to incorrect imaging of morphology as can be seen, for example, in Fig. 9.”

A clear description of the drawbacks of this method (presumably a small field of view, photobleaching due to the relatively large # of images required, slow correction due to the large number of images and thus worsened performance in moving samples, etc) would add value to the discussion. Even better if the strengths/weaknesses of the method are compared against the direct wavefront sensing 2P method as this method is generally considered state of the art.

We now include direct wavefront sensing, and we discuss and demonstrate the relative advantages of sensorless AO, 1p and 2p direct wavefront sensing. Clearly 2p wavefront sensing works very well but it is also adds considerable cost.

In the conclusion, we now discuss the relative merits of different wavefront sensing approaches.

“Here we have presented the combination of structured illumination microscopy with adaptive optics. While the improvement with Adaptive Optics is strongly sample dependent, the results can be substantial. We have demonstrated a lateral resolution of 148 nm and an axial resolution of 578 nm at a 670 nm emission wavelength when imaging through the body of an adult *C. elegans* at a depth of $\sim 80 \mu\text{m}$. The improvement in axial resolution is at least 50% and the improvement in SNR is more than 70%. Not only do the resolution and SNR improve, but, more importantly, the image fidelity improves as well. Under severe conditions, such as blurry images of AJs in *C. elegans*, confocal illumination can be used to assist the image quality assessment, removing the dependence on the object in the metric function. We have also implemented 3DSIM with direct wavefront sensing with confocal illumination. We used the confocal spot as the “guide star.” We demonstrated that the widefield image based sensorless AO method, the confocal spot based sensorless AO method, and confocal illumination based direct wavefront sensing AO method can all help to improve the image quality and fidelity of the 3D-SIM images. Each approach has strengths and weaknesses, and they can further be applied in conjunction.”

Would be useful to describe in more detail what the arrows represent in Fig. 2a, in the caption – this appears to be the polarization vectors applied in illumination (and how they are generated) but the casual reader may not appreciate this.

We now provide a brief description in the caption.

“PG: pattern generation. The polarization direction indicated by arrows is shown after each element for both on (1) and off (0) pixels.”

Is there a reason that the maximum illumination spatial frequency was not used for the SIM, as in previous SR-SIM papers? Presumably this would result in a greater resolution improvement, in principle.

The illumination spatial frequency is placed at $\sim 80\%$ of the pupil. Gustafsson used a spatial frequency of 92% of the pupil. Using a slightly lower spatial frequency improves the performance of the SIM reconstruction algorithm which is helpful in aberrating conditions (although we are correcting the aberrations, the correction is not perfect). We note that recent SIM systems have used similar resolution enhancements as ours. Markwith et al. used a resolution enhancement of 1.8 (*Video-rate multi-color structured illumination microscopy with simultaneous real-time reconstruction*, Nat. Comm. **10** 4315, 2019)

How much greater is the intensity (or illumination power) for the ± 1 orders vs. the zero order beam?

We now provide this information. The +/-1 orders are roughly 3x more intense than the 0 order.

The fits in Fig. 5e don't fit the experimental points particularly well. This makes me wonder if the parabolic assumption is in fact an oversimplification, and if better results might be obtained with a more complex model/fit. Please clarify.

The parabolic fits to the metric vs. aberration amplitude are now shown in Fig. S3. The reviewer is correct that the parabolic fit is an approximation that is only accurate over a sufficiently small range. A more sophisticated model would indeed be an interesting area to explore and some work has been done in this area (Booth, Opt. Lett. 32(1), pp. 5-7, 2007). We have found that the parabolic fit works fairly well and a second iteration can be used to improve the convergence and overcome the errors with the parabolic fit.

To address the question of larger aberrations where the sensorless approach does not work, we now use direct wavefront sensing.

In Fig. 6d, the image looks overall weaker in intensity than Fig. 6c, is this due to photobleaching? In fact it looks like features in Fig. 6c are actually missing in Fig. 6d. Thus, I am not convinced that correcting the sample aberration offers a significant advantage over the static system correction. I also am not convinced that the image in Fig. 6c is significantly fuzzier than in Fig 6d – in fact it looks sharper (which might be due to noise). It's also not clear that the image 'looking smoother' is an indication that Fig. 6d is 'better' than Fig. 6c as the authors imply, i.e. if higher spatial frequency components are restored it might in fact be that the fully corrected image looks less smooth.

We respectfully disagree. The area in the dotted white square was the area where the correction was performed. In this region the images after correction show more detail, the structures are thinner and brighter, and the SNR is higher. I believe the improvement in SNR is especially evident in the axial cross sections. Perhaps 'smoother' is the wrong term, but after aberration correction, the morphology is more apparent.

We now show a second image of the rice blast fungus, Fig. 9, in which the correction is done using direct wavefront sensing. In the axial cross sections in this figure, it is also very clear that the correction makes the morphology more apparent.

Reviewer #2 (Remarks to the Author):

Lin et al present adaptive optics (AO) applied to 3D structured illumination microscopy, and its application to various samples. They show that AO can be of great benefit to SIM reconstructions, both in suppressing artifacts, but also in obtaining high spatial resolution. The AO uses iterative optimization of Zernike modes using an image metric. In addition, the illumination can be patterned to add "structure" to the sample to help the algorithm converge, which is inspired by the IsoSense paper (Žurauskas, M., et al., Optica, 2019. 6(3): p. 370-379.).

In contrast to other publications that have combined SIM and AO, the authors claim that this is the first application of 3D SIM (three beam interference and 3D data acquisition) and that it goes

further than the IsoSense publication in terms of the complexity of the aberrations. To my understanding, the IsoSense microscope is also a 3D SIM type, but I agree with the authors that in that publication, no serious aberrations were corrected (it looked more like correction of systems aberrations). In that sense, the authors' publication shows indeed more of the potential of AO and 3D SIM and the detailed results can serve as a reference what kind of gains can be expected. As such I found the manuscript interesting, albeit a bit incremental on previous publication.

These are the first results in which 3D SIM with AO is demonstrated, and the improvement to the resolution in the axial direction is significant (as in Fig. 9). So we don't agree that our work is incremental. The IsoSense system is capable of 3D SIM but it is not demonstrated. There are not axial cross-sections in the IsoSense paper demonstrating the axial resolution.

We have made substantial changes to the paper that we hope will convince the reviewer that our results are significant. We now demonstrate three different modes of AO correction: sensorless AO over a region, sensorless AO with confocal illumination, and direct wavefront sensing using a Shack Hartmann Wavefront Sensor and confocal guide star.

larger issues:

The authors claim that they can correct large aberrations. The largest aberrations occur for the c-elegans, however they are of low complexity (it is mainly dominated by astigmatism). The magnitude is still lower than one wavelength, so it can also be debated if this is large. I am not fully convinced that the iterative scheme can deal well with large and complex aberrations, as the starting image will be of low SNR and contrast, and the sensitivity to the aberration probing will become low. For large aberrations, a Shack Hartmann sensor might be more robust, as it does not suffer from decaying image contrast (or other image metric). I understand that a Shack Hartmann comes with its own limitations, but I am not convinced that the iterative sensorless scheme is superior. To my knowledge, Zernike mode based sensorless AO has only been shown for relatively benign aberrations that were neither complex nor too far from the aberration free case. So I invite the authors to reconsider the phrasing of the magnitude and complexity of aberration correction.

Another concern is that the authors only do aberration corrections for relatively basic, low order aberrations (coma, astigmatism, spherical). This essentially prevents them from correcting any more complex, higher order aberrations. In other AO work, much higher aberrations modes have been considered (Wang, K., et al. Nature Communications, 2015.). This leaves the concern that the authors picked some examples where this low complexity wavefront correction works well (i.e. correction of Astigmatism on the c-elegans). Would it be possible to make an experiment with more complex aberrations? Maybe using a phantom to introduce higher order aberrations? Otherwise some claims need to be revised.

We now demonstrate both sensorless AO and direct wavefront sensing with a Shack-Hartmann Wavefront Sensor. With sensorless AO it is true that we only correct 5 modes in most cases, but that does not mean that the aberrations are small. We also point out that aberrations on the order of 1 wavelength are large and cause significant distortion. In supplementary figure 1, we show

the effect of different aberrations on the PSF, and we believe that it is clear that an aberration of ~ 1 wavelength has a strong effect on image quality.

Further, we now show results with direct wavefront sensing. We measure the wavefront with 150 degrees of freedom and set the wavefront with 69 degrees of freedom. This means we can control the first 20 Zernike modes (up through the 5th order) with amplitude of greater than 1 micron.

Another interesting point is that the aberration loop runs separately for each aberration mode and then at the end the ideal coefficients are computed. Would it make more sense to update each mode once it has been tested, such that subsequent modes start with less overall aberrations (and hence higher SNR and contrast)? My intuition is that with such a procedure, and the addition to run the whole optimization multiple times, sensorless AO could pull itself out of a bit more complicated aberrations.

The wavefront is updated after iterating over each Zernike mode. We have changed the text in the methods section, "Sensorless AO," to make this clear. We could indeed correct higher order aberrations with more sensorless iterations, but we have tried to optimize the correction while keeping the number of iterations low.

"The peak of the parabolic fit is set as the optimal correction strength for the corresponding Zernike mode and the mirror shape is adjusted accordingly. This procedure is repeated on all modes to be corrected to get the optimal strength for each mode separately."

For the Neuronal imaging, the authors applied a dot like pattern for illumination. Does the mask for 3D SIM need to be removed for that or how is the pattern generated? The authors should also show a Fourier transform with and without the dot pattern applied to better illustrate how it can increase the frequency content. As a further question, do the authors assume that the dot pattern is not distorted by the aberrations? Can that be further justified?

We now use a single confocal spot to increase the high-frequency content in the image. The confocal spot is shown in Fig. 7. The confocal spot is generated using a different excitation path that bypasses the pattern generation optics (SLM, mask, etc.). The confocal spot will of course be distorted by the aberrations. With sensorless AO, as the wavefront is corrected, both the confocal spot and the peak fluorescence generated by the confocal spot will improve leading to an overall increase in the metric. With direct wavefront sensing, the confocal spot is being imaged by multiple low-NA lenslets, so the distortion of the confocal spot does not affect the performance of the Shack-Hartmann Wavefront Sensor (see Tao et al. Opt Expr. 20(14) pp. 15969-15982, 2012).

Smaller points:

There are different definitions for numbering Zernike modes. I assume the authors use the Noll index. This has to be more clearly defined. Likewise, the authors are a bit imprecise about their AO algorithm. What is the step size ($-2aa, -aa, 0, aa, 2aa$)? Is this in microns? Please give this number also in radians, to give a better idea how this scales to the wavelength.

Yes. We use the Noll ordering. We have now clarified this in the text. ‘a’ is a parameter that indicates the added aberration in the sensorless AO measurement. We specify in the text that ‘a’ is set to either 0.1 μm or 0.2 μm .

In the first paragraph under heading *Image-based Sensorless AO-3DSIM*

“The Fourier Transform (FT) of the 3D-SIM image (insets) illustrates the aberrations clearly in the form of a cross shape in each OTF copy, revealing a dominating aberration of astigmatism (Zernike mode 5, Noll ordering [39]).”

It is not clear what the values on the wavefront sent to the mirror are (e.g. Figure 3i, 4f 5f 6h, microns? Radians?). Scaling it to 2π radians might be most instructive.

We have now modified the figure captions to specify the units for all wavefront measurements.

Table 1: any explanation why the gain in axial resolution in SIM with AO is so much more dramatic in the z-direction? In widefield with and without AO, the gain is rather small.

We now discuss this important result in the conclusion. Many aberrations lead to a PSF that, in the lateral direction, retains a central lobe that is not much wider than the unaberrated PSF central lobe. However, the axial extent of the PSF is increased significantly. This can be seen in spherical aberration in which intensity is removed from the central lobe but its lateral extent is essentially the same. However, the the PSF is broadened in the axial direction. There is an interesting discussion of the effect of random phase screens on the PSF in chapter 8 of *Statistical Optics* by Goodman (Wiley 2000). The random phase screen creates a broad pedestal upon which the core of the PSF ‘sits’, but the core is not broadened.

“From our work, we see that the improvement to axial resolution in 3D-SIM is substantial. The axial resolution degrades more quickly with aberrations. This can be seen by examining the effect of aberrations such as coma and spherical aberration on the PSF; these modes do not change the lateral size of the PSF central lobe although they may add side lobes and reduce the main lobe intensity, but they clearly increase the axial extent of the PSF.”

The authors compute signal to noise, although it is not clear to me how this is done. The text reads: “The noise is defined as the standard deviation (STD) around the maximum, which is defined as the signal. The image SNR.” Is this performed in real or reciprocal space? The only related method I know is measuring standard deviation in frequency space outside of the OTF and comparing it to the DC frequency component. The authors have to better explain their method.

We have now clarified how the SNR is calculated. We have added the following section to the Methods:

“SNR Calculation

To calculate the SNR, we first crop an ROI around an identifiable object (bead, nerve fiber, etc.) surrounded by a relatively dark background. We calculate the mean value of the pixels forming the object and divide by the STD of the surrounding area which is defined as the pixels with values lower than a threshold determined by the adaptive threshold algorithm in the python scikit image package (`threshold_local()` in `skimage.filters`, https://scikit-image.org/docs/0.18.x/api/skimage.filters.html#skimage.filters.threshold_local).”

Measuring the standard deviation in frequency space outside the OTF would only yield the camera noise in a widefield image. In the SIM images, the region outside the effective OTF is set to 0.

Overall, enthusiasm for this manuscript is high, but I recommend strengthening and clarifying the manuscript.

We thank the reviewer for his positive words.

Reviewers' Comments:

Reviewer #1:

Remarks to the Author:

The authors have improved their manuscript, have addressed most of my concerns, and the inclusion of more methods of wavefront sensing in conjunction with 3D SIM will be of interest to the field. I have several remaining concerns I would like to see addressed before publication:

-The use of the confocal spot is an interesting way to implement direct wavefront sensing, but I would imagine that the pinhole would filter out some of the modal wavefront structure necessary for wavefront measurement, a concern also noted by Wang and Betzig, 2014 Nature Methods. Presumably this effect is a strong function of the pinhole diameter. Please comment on this issue and discuss how the pinhole aperture was chosen.

-The authors state that the thickness of the *C. elegans* worm was ~80 μm and the rice sheath was ~60 μm . Were these verified experimentally? A bit of literature searching (and personal experience) reveals that at least for *C. elegans* adults there is considerable difference in thickness in *C. elegans* adults, with young adults being closer to 50 μm in diameter. Please make very clear in the text if the thicknesses were educated guesses or if they are actual measured values: AO through 80 μm is different than AO through 50 μm .

-I appreciate the bleaching measurements provided in SI Fig. 6. However, I am unclear how many iterations were used in the datasets in each figure, and would advise clarifying this in the paper. The number of iterations seems to have significant impact on the bleaching, e.g. by iteration 2 there is obvious fluorescence loss and by iteration 4 more than half the fluorescence is lost. This is not 'minor photobleaching', although I agree that within one iteration bleaching appears to be minor. Please clarify these points.

-Similarly, I appreciate the frame #s in SI Table 2, but think it is important to mention explicitly in the Discussion section that - at least for the methods reported in this paper - the time for indirect AO correction is a substantial fraction of the time required to acquire the data. Similarly, it will be useful to report the amount of time required to process the data and actually obtain the correction once the data has been gathered. As is, I am not convinced the indirect method is really suitable for imaging extended temporal dynamics - providing an honest assessment of the total time (including computation) required for correction will both help biologists assess if the method 'works' for their dynamics of interest and provide tool developers the impetus to build on this work.

-I am still not quite convinced of the biological importance of this technique. While I agree that SNR and resolution are improved after AO, the descriptions of the biological structures in many of the figures are suitably vague that the authors do not make as convincing a case as they could. I think this could be improved by describing what we are seeing after AO correction more clearly. In the interest of constructive criticism:

a) In Fig. 5e vs. 5c, why is Fig. 5e better and what can be seen more clearly after AO correction in Fig. 5e that is not obvious in Fig. 5c? The uncorrected region actually looks dimmer and less defined in Fig. 5e than Fig. 5c - thus I am not convinced of the authors' assertion that a 'general improvement can be seen'. In the corrected region what am I supposed to observe in the dotted region that I am not seeing in Fig. 5c? I suspect the reader who is not an expert on rice sheath cells will be similarly bemused. The authors claim that Fig. 5e is 'smoother and more continuous' but again the visual impression is that 5e actually shows less than 5c. Perhaps a higher magnification view would help here, and better description of what we 'should' see in such samples.

b) Similarly in Fig. 6d,e the authors claim 'more fidelity' after AO correction and that the 'isolated nerve fibers can be clearly recognized as continuous'. Without clarifying to the reader what 'fidelity' means for this structure, the first statement is meaningless; and for the second statement the 'upper' fiber actually looks less continuous after AO correction. Perhaps including an arrow can better guide the reader to what you are describing here.

c) Fig. 8d, I agree these structures appear sharper - but what am I looking at?

d) Fig. 9b, c - presumably the point is that the 'hollow' structure of the hypha is what is revealed after AO correction, i.e. this is the 'incorrect morphology' in the aberrated image noted in the

Discussion. Is that correct? Guiding the reader to the point would again be helpful, especially since they are likely not familiar with the 'correct' morphology.

-Table 1 is missing standard deviations and N, please report these so the numbers are statistically meaningful.

-Are different lateral/axial scales used in Fig. 4d, and what are the scales in question?

Reviewer #2:

Remarks to the Author:

I have received and studied the detailed and comprehensive revisions by Lin et al. My concerns were thoughtfully addressed in their rebuttal. In the process, quite large additions were made to the paper, which makes the study much more complete: it now compares three different modalities for adaptive optics, and their application to 3D SIM. This goes along with additional samples that were imaged.

While I applaud the efforts by the authors, the addition of new technologies, ideas and material also necessitates more reviewing. Thus I hope the author's are not discouraged by the concerns raised below, but can see it as an encouragement that their new ideas were considered exciting and thought-provoking.

Most importantly, the new confocal laser spot guide star, filtered with a pinhole, appears to be a novel concept, as it breaks with the notion that one needs a nonlinear (two-photon) process to generate a well isolated spot for wavefront sensing. This would be a welcome change. However, some concerns arose for this. It appears that a judicious choice for pinhole 2 must be made, such that it rejects out-of-focus blur, but does not truncate too many aberrations modes. It was not clear how this choice was made, and how it influences the results. Only a dimension was given (500 microns, for which the corresponding Airy unit should be given too).

Also some discussion on how this compares to the published 2 photon guide star work by the Betzig lab should preferably be included. As another difference, previous art scanned and descanned the guidestar, which is claimed to smooth/average the encountered aberrations. Furthermore, I can see a benefit in scanning, namely that it is easier to get a fluorescence signal if labeling is not uniform.

Further concerns arose when the confocal laser spot is used for sensorless AO: The spot itself will get aberrated by the sample, and the iteration of the wavefronts on the DM, both on the way into the sample, and on the way back. Are those forward and backward aberrations problematic? One would need to assume that there is no dispersion in the sample, else the forward (laser) and backward (fluorescence) aberrations would be different, and the correction algorithm would not necessarily converge to the desired correction (fluorescence aberrations).

But beyond that, trying to solve the forward and backward problem at once was always treated with some restraint/concern in the field. Thus I would like to hear the authors' thoughts on this. Off note, in the Direct wavefront sensing scheme, it is in an approximation still a single pass measurement. The WFS sensor has such low magnification on each element that some aberrations on the spot will not matter. In my view there it should not be a problem that the ingoing beam is getting aberrated as well (DM + sample).

A question for the Sensorless AO implementation (without confocal guide star): How was the sample illuminated? With a single SIM pattern, a rapid sequence of SIM patterns, or WF?

I found it interesting that the sensorless AO performed better under severe aberrations than direct wavefront sensing. I would have assumed that sensorless AO, when buried deep in aberrations, would struggle to dig itself out. I would assume one can design a Shack-Hartmann sensor in many ways, could it be that the one of the authors was designed towards small magnitudes of aberration modes (and as such was more prone to large changes)?

On the biological images (details below), I have for some data difficulties appreciating improved resolution after AO correction. Some line profiles show finer bumps after applications of AO, others the opposite. But line profiles can also be fickle and be prone to picking a suitable one. Could the authors apply Fourier Ring Correlation or image decorrelation* (which can work on single image frames in contrast to FRC) to assess resolution on biological samples? In terms of resolution gain, the beads show a clear gain, and there quantitative values are given, which was appreciated.

*Descloux, A., Kristin Stefanie Großmayer, and Aleksandra Radenovic. "Parameter-free image resolution estimation based on decorrelation analysis." *Nature methods* 16.9 (2019): 918-924.

Further questions:

What is the purpose of the red laser? Is it used to calibrate SWHS sensor and mirror?

Figure 8g: is the inset showing the spot before or after correction?

Figure 8b-c: The resolution does not seem to get higher upon visual impression, but the signal goes up. I cannot see finer features. A bit a similar comment about the data shown in Figure 6-7. It is hard to see finer features, but signal went up. Can the authors please comment on this (or to estimate the resolution with a metric like Image Decorrelation)?

The insets in Figure 9 B and C, do they show the same axial cross sections? They look structurally quite different. Also in Figure 9c, system correction shows a prominent structure (looking like a diagonally running bar) that is gone in the sample correction. Can the authors please comment on this?

Overall, I think this is important work, it demonstrates a comparison between different AO schemes applied to 3D SIM for the first time and it might introduce a more affordable concept for guide stars.

March 18, 2021

Manuscript NCOMMS-19-1472556-T, “Subcellular three-dimensional imaging deep through multicellular thick samples by structured illumination microscopy and adaptive optics” by Lin et al.

RESPONSE TO REVIEWER COMMENTS

We thank the reviewers for their comments and appreciate that they both have positive opinions of the manuscript. We have modified the manuscript and added to the supplemental information to address their comments. We have clarified the discussion of the pinhole and its effect on the aberration measurement (Fig. S8). We have redone Figure S7 to quantify the bleaching. We have added an additional part to figure S6 to explain the behavior of the Shack Hartmann Wavefront Sensor. We have made additional measurements to quantify the improvement with SI and AO (Fig. S4 and supplementary tables 3, 4 and 5). We have tried to address their concerns on the biological importance of the technique and the visual improvement in the images. We feel that the use of AO makes a substantial improvement in the 3D-SIM images, and we hope that we have satisfied the reviewers that this is the case.

Below, we address the specific comments from the reviewers. Their comments are in black, our responses are in blue, and changes to the text are in red.

Reviewer #1 (Remarks to the Author):

The authors have improved their manuscript, have addressed most of my concerns, and the inclusion of more methods of wavefront sensing in conjunction with 3D SIM will be of interest to the field. I have several remaining concerns I would like to see addressed before publication:

We thank the reviewer for his positive opinion of our manuscript. Below we try to address the remaining concerns.

-The use of the confocal spot is an interesting way to implement direct wavefront sensing, but I would imagine that the pinhole would filter out some of the modal wavefront structure necessary for wavefront measurement, a concern also noted by Wang and Betzig, 2014 Nature Methods. Presumably this effect is a strong function of the pinhole diameter. Please comment on this issue and discuss how the pinhole aperture was chosen.

We apologize for not providing more details on the pinhole before. The effect of pinhole size in AO has been discussed by Tao et. al (Opt. Lett. 36(17) pp. 3389-3391, our ref. 32). We now add a discussion and accompanying figure in the supplementary information, Fig. S8, and add the supplementary reference Opt. Expr. 20(14) 15969.

The pinhole has an 8.3-micron diameter at the sample plane, significantly bigger than the point spread function. The main purpose of the pinhole in our setup is to block some out-of-focus light. We tried several different pinhole sizes and chose a conservatively large pinhole. The pinhole effectively blocks light +/- 5 microns away from the focal plane. In Supplementary Figure S8c, we now show plots of the wavefront error measured with and without the pinhole vs. the wavefront set on the DM for different Zernike modes. These plots show that the pinhole only begins to cause problems when the RMS wavefront amplitude exceeds ~0.5 μm .

The figure above (now Fig. S8b) shows the integrated intensity through the pinhole from a PSF as a function of defocus without wavefront error and then with approximately 2 wavelengths of wavefront error (indicated in radians in the figure).

The size of the pinhole image on the deformable mirror is an Airy Disk with central lobe diameter of 0.6 mm, much less than the actuator pitch of 1.5mm. This indicates that the pinhole size does not affect the measurement of the wavefront at the scale that can be corrected by the DM. The pinhole and the deformable mirror are respectively in the front and back focal planes of a 250mm focal length achromat. The deformable mirror actuator pitch is 1.5mm and, for small actuator throw, the DM can be thought of as a grating with grating wavelength 3mm. For $\lambda=500\text{nm}$, the DM will deflect light at a maximum angle of $\sim 2\text{e-}4$ which corresponds to a deflection of ~ 50 microns at the pinhole location. Therefore, for small wavefront errors the, pinhole has no effect. For larger wavefront errors, the pinhole will begin to affect the results. This is because the frequency content of the beam will increase as the wavefront error increases – more terms must be kept in the expansion of $\exp(-j\varphi(\vec{r}))$ as $\varphi(\vec{r})$ increases.

We include the following text in the Figure S8 caption

Following the analysis in in sec. 2.7 of Tao et al.³, a pinhole of size λ/d_{sub} will attenuate spatial frequencies above $1/2d_{\text{sub}}$. For our wavefront sensor, the lenslet pitch is 150 microns. With $\lambda = 509 \text{ nm}$, this corresponds to a pinhole size of 3.4 milliradians. At the position of PH2, this corresponds to 288 microns. Therefore, a pinhole of 500 microns should not limit the measurement of frequencies that can be measured by our wavefront sensor.

The size of the pinhole image on the deformable mirror is an Airy Disk with central lobe diameter of 0.6 mm, much less than the actuator pitch of 1.5mm. This indicates that the pinhole size does not affect the measurement of the wavefront at the scale that can be corrected by the DM. The pinhole and the deformable mirror are respectively in the front and back focal planes of a 250mm focal length achromat. The deformable

mirror actuator pitch is 1.5mm and, for small actuator throw, the DM can be thought of as a grating with grating wavelength 3mm. For $\lambda = 500 \text{ nm}$, the DM will deflect light at a maximum angle of $\sim 2e-4$ which corresponds to a deflection of ~ 50 microns at the pinhole location. Therefore, for small wavefront errors the, pinhole has no effect. For larger wavefront errors, the pinhole will begin to affect the results. This is because the frequency content of the beam will increase as the wavefront error increases – more terms must be kept in the expansion of $\exp(-j\varphi(\vec{r}))$ as $\varphi(\vec{r})$ increases. As we show in Fig. S8(b), the pinhole in our setup does not begin to affect the wavefront measurement until the RMS amplitude exceeds $\sim 0.5\mu\text{m}$.

-The authors state that the thickness of the *C. elegans* worm was $\sim 80 \mu\text{m}$ and the rice sheath was $\sim 60 \mu\text{m}$. Were these verified experimentally? A bit of literature searching (and personal experience) reveals that at least for *C. elegans* adults there is considerable difference in thickness in *C. elegans* adults, with young adults being closer to $50 \mu\text{m}$ in diameter. Please make very clear in the text if the thicknesses were educated guesses or if they are actual measured values: AO through $80 \mu\text{m}$ is different than AO through $50 \mu\text{m}$.

The *C. elegans* in Fig. 4 was an adult more than 3 days old and had reached its largest size with a diameter of $\sim 80 \mu\text{m}$. There is no doubt that the images in Fig. 4 prove the AO improvement at a depth of $\sim 80 \mu\text{m}$ through the *C. elegans*. The DIC image in Fig. 4 can also illustrate the worm diameter from another perspective. As we can only see less than half of the worm in the field of view, which has a size of $45.57 \mu\text{m} \times 45.57 \mu\text{m}$, the worm radius is definitely no less than $40 \mu\text{m}$. Note that we mounted the worm very carefully and made sure that the worm was not squeezed and can move freely. The *C. elegans* used in Fig. 6, 7, 8, 10, are all about ~ 3 days old and have a diameter of $45\text{--}48 \mu\text{m}$ at the imaging position. The adherens junctions are roughly in middle of the worm, and the RIC interneurons are somewhat more than halfway through the worm body.

The rice sheath sample is provided by Dr. Khang. The details about the sheath thickness and the approach to consistently control the thickness of the trimmed sheath can be found in the reference (see Notes 4): Jones K, Khang CH. Visualizing the Movement of Magnaporthe oryzae Effector Proteins in Rice Cells During Infection. In: Plant Pathogenic Fungi and Oomycetes: Methods and Protocols (eds Ma W, Wolpert T). Springer New York (2018). In the text, we indicate that the sample is $\sim 60 \mu\text{m}$ thick.

We have changed the text on page 9 from

For live-cell imaging, we used hand-cut optically clear rice sheath tissue ($\sim 60 \mu\text{m}$ thick), consisting of a layer of epidermal cells, within which the fungus resides, and a few underlying layers of mesophyll cells (Fig. 5a).

To

For live-cell imaging, we used hand-cut optically clear rice sheath tissue consisting of a layer of epidermal cells, within which the fungus resides, and a few layers of mesophyll cells (Fig. 5a). The sample is approximately $60 \mu\text{m}$ thick, and we are imaging 30 hours after inoculation when the fungus is approximately $20 \mu\text{m}$ below the sample surface.

We have added the following sentence to the paragraph on page 14 before Fig. 9.

In Fig. 9, we are imaging 48 hours post inoculation, and the fungus is approximately 30 μm below the sample surface.

-I appreciate the bleaching measurements provided in SI Fig. 6. However, I am unclear how many iterations were used in the datasets in each figure, and would advise clarifying this in the paper. The number of iterations seems to have significant impact on the bleaching, e.g. by iteration 2 there is obvious fluorescence loss and by iteration 4 more than half the fluorescence is lost. This is not 'minor photobleaching', although I agree that within one iteration bleaching appears to be minor. Please clarify these points.

The iterations referred to in SI Fig. 6 (now supplementary Fig. S7) are separate experiments, so the difference in intensity between these iterations is not relevant. To clarify this, we now use the term "trials" instead of "iterations". We further add an exponential fit to each trial so that we can estimate the amount of photobleaching.

Based on these fits, the amount of photobleaching (that is the drop in intensity) that occurs during sensorless AO wavefront correction which requires 25 images, is in the range 2.2% to 4.4%. To acquire a set of raw 3D SI data requires ~390 images, and the photobleaching is in the range 30% to 50%. We have added to the discussion in the caption to Fig. S7 to clarify this point.

For a typical number of sensorless AO iterations, 25, the amount of photobleaching is ~ 2.2% to 4.4%. For an AO-3DSIM stack with 390 raw images, the amount of photobleaching is ~ 30% to 50%.

-Similarly, I appreciate the frame #s in SI Table 2, but think it is important to mention explicitly in the Discussion section that - at least for the methods reported in this paper - the time for indirect AO correction is a substantial fraction of the time required to acquire the data. Similarly, it will be useful to report the amount of time required to process the data and actually obtain the correction once the data has been gathered. As is, I am not convinced the indirect method is really suitable for imaging extended temporal dynamics - providing an honest assessment of the total time (including computation) required for correction will both help biologists assess if the method 'works' for their dynamics of interest and provide tool developers the impetus to build on this work.

We have added a paragraph to the discussion section, and now discuss how long it takes to both acquire the raw SI data and to perform the wavefront correction. We disagree that the indirect method is not suitable for imaging temporal dynamics. While the correction may not be instantaneous, it is not necessary to correct before every frame. Therefore, we feel this method can still be appropriate for imaging dynamic events. We would also point out that sensorless AO requires 25 images, and a typical 3D SIM stack requires 390 images. So, the time for sensorless AO is relatively modest.

The time required for AO correction is 6 sec., and the time to acquire the 3D-SIM raw data is ~90 sec. although these times can be reduced with further optimization of the microscope control software (Supplementary Table 2 contains the details of each acquisition.) Therefore, the time for AO correction is relatively modest compared to the time to acquire the SIM data. Furthermore, when correcting time-lapse data, the AO correction must not necessarily be performed before every image stack. Typically, the wavefront errors are caused by structures that are relatively static even during live imaging. The acquisition time can also be decreased for live imaging through the use of shorter exposure times, and, as has previously been noted, the time resolution is not limited by the time to acquire one stack but rather by the time to acquire one depth-of-focus ¹⁴.

-I am still not quite convinced of the biological importance of this technique. While I agree that SNR and resolution are improved after AO, the descriptions of the biological structures in many of the figures are suitably vague that the authors do not make as convincing a case as they could. I think this could be improved by describing what we are seeing after AO correction more clearly. In the interest of constructive criticism:

As the reviewer points out, SNR and resolution are improved after the AO correction. This by itself is important in microscopy and indicates that we have indeed improved the final images. In optical imaging, aberrations always degrade the optical transfer function. There is no aberration that will make the image brighter but incorrect. Therefore, if the corrected image is brighter, that should give confidence that the AO has improved the fidelity of the image to the underlying structure.

We would also point out that the correction of beads in Fig. 4 shows a clear improvement with AO. While we agree that imaging beads is an artificial case, there is no dispute about what a bead should look like; so the improvement is clear. If the improvement in this case is clear, that should give confidence in the AO improvement of the other images. Given the difference between images of beads with 3DSIM and AO compared to widefield without AO, we feel that the advantages of the combination of AO and 3DSIM when imaging biological structures is evident. While we agree that the benefits of AO are not always immediately obvious, this is not a reason to discard the results. We feel that AO correction is an important step.

We now try to make the improvement to the biological images clearer in the text and figures by explaining more clearly what some of these structures should look like so that the improvement is clear. Below, we address the specific comments from the reviewer.

a) In Fig. 5e vs. 5c, why is Fig. 5e better and what can be seen more clearly after AO correction in Fig. 5e that is not obvious in Fig. 5c? The uncorrected region actually looks dimmer and less defined in Fig. 5e than Fig. 5c - thus I am not convinced of the authors' assertion that a 'general improvement can be seen'. In the corrected region what am I supposed to observe in the dotted region that I am not seeing in Fig. 5c? I suspect the reader who is not an expert on rice sheath cells will be similarly bemused. The authors claim that Fig. 5e is 'smoother and more continuous' but again the visual impression is that 5e actually shows less than 5c. Perhaps a higher magnification view would help here, and better description of what we 'should' see in such samples.

We appreciate the comment. As the reviewer suggested, we revised some panels in Fig. 5 to show ER network arrangements in a higher magnification view. This revision turns out to be very helpful to highlight the improvement after AO correction. In filamentous fungi, the ER can be seen as the spherical perinuclear ER and the reticulate peripheral ER along the hypha at the fluorescence light microscopy level (Boenisch et al., 2017). The revised Fig.5e clearly shows smoother and more continuous perinuclear ER (arrowheads) and peripheral ER (arrows), which look fuzzy in Fig. 5c. Also, there is substantial improvement in the axial resolution after AO correction. The z slice image in Fig. 5e shows a ring-shaped EGFP fluorescence pattern, demonstrating the spherical organization of perinuclear ER, which is not resolved in Fig. 5c.

We have modified figure 5 and replaced the text

With AO correction and SIM, we can clearly identify the integral structure of the ER and their distribution around the nuclei in all three dimensions as shown in Fig. 5e. While, in Fig. 5c, due to the aberration and its induced reconstruction artifacts, the image looks fuzzy, and the ER structure is not smooth and continuous as it is in Fig. 5e.

with

In filamentous fungi, the ER can be seen as the spherical perinuclear ER and the reticulate peripheral ER along the hypha at the fluorescence light microscopy level (Boenisch et al., 2017). The revised Fig.5e clearly shows smoother and more continuous perinuclear ER (arrowheads) and peripheral ER (arrows), which look fuzzy in Fig. 5c. Also, there is substantial improvement in the axial resolution after AO correction. The z slice image in Fig. 5e shows a ring-shaped EGFP fluorescence pattern, demonstrating the spherical organization of perinuclear ER, which is not resolved in Fig. 5c.

b) Similarly in Fig. 6d,e the authors claim 'more fidelity' after AO correction and that the 'isolated nerve fibers can be clearly recognized as continuous'. Without clarifying to the reader what 'fidelity' means for this structure, the first statement is meaningless; and for the second statement the 'upper' fiber actually looks less continuous after AO correction. Perhaps including an arrow can better guide the reader to what you are describing here.

As can be seen in the cartoon in Fig. 6, there are two RIC interneurons that travel towards the posterior of the worm from the pharynx. In Fig. 6d, it appears that there are three fibers, and in fig. 6e there are only two. In the original figure, the intensity is not that bright throughout the image so one could argue that the structure is not more continuous in the corrected image. We now adjust the scaling so that the continuity of the neuron in fig. 6e is more evident. We have removed imprecise language such as 'more fidelity' throughout the text.

We have modified Fig. 6 to include arrows.

In the text, we have changed

We took a 4 μm stack along the z-axis with a step of 0.2 μm for 3D-SIM, focusing on the axon as shown in Fig. 6. The effect of AO correction can be seen by comparing Figs. 6b and 6c in widefield and Fig. 6d and 6e in 3D-SIM. The filament structure is clearer, and the signal intensity becomes stronger after AO correction. With AO correction, the image of the nerve fiber, Fig. 6e, has better resolution and more fidelity compared to the dim and noisy image without AO correction, Fig. 6d. In y-z slices (insets), the peak intensity after AO correction is higher than before AO correction, while the noise level is much lower. The isolated nerve fibers can be clearly recognized as continuous in the image.

To

The structural details of the RICR/L neurons are illustrated in the graphic rendition shown in Fig. 6a. We took a 4 μm stack along the z-axis with a step of 0.2 μm for 3D-SIM, focusing on the axons before the ring as the image overlay in Fig. 6a shows. The effect of AO correction can be seen by comparing Figs. 6b and 6c in widefield and Fig. 6d and 6e in 3D-SIM. As can be seen in Fig. 6a, there are two RIC interneurons that travel towards the posterior of the worm from the pharynx. In Fig. 6d, there appear to be three filaments along the x-direction. After AO correction, Fig. 6e, it is clear that there are only two fibers as indicated by the arrows. In the y-z slices showing the fiber cross sections (insets), the peak intensity after AO correction is higher than before AO correction, while the noise level is much lower. The isolated nerve fibers can be clearly recognized as continuous in the AO corrected image.

c) Fig. 8d, I agree these structures appear sharper - but what am I looking at?

We are looking at *ajm-1::GFP* expressed in the adherens junctions. The adherens junctions form into bundles between the epithelial cells. As we state in the text, we are imaging the anterior bulb of the pharynx where the adherens junctions between cells presumably function to maintain structural integrity during pharyngeal pumping. These structures can be seen at low resolution in Figure 4 of our reference 46 (Raharjo et al.). We have not been able to find an image in the literature at comparable resolution to our Figure 8.

We have modified the Fig. 8, providing zoomed-in view of these adherens junctional structures. And we have also changed the text on the top of page 13 from

To demonstrate the direct wavefront sensing method, we imaged the AJs inside the posterior bulb, as shown in Fig. 8a. By comparing the 3D-SIM images before (Fig. 8b) and after (Fig. 8c) the sample aberration correction in all three dimensions, we can see clearly that fine structural details become sharper and better resolved.

to

To demonstrate the direct wavefront sensing method, we imaged bundles of AJs inside the posterior bulb, as shown in Fig. 8a. In the 3D-SIM image after sample correction, Fig. 8c, these bundles appear sharper.

d) Fig. 9b, c - presumably the point is that the 'hollow' structure of the hypha is what is revealed after AO correction, i.e. this is the 'incorrect morphology' in the aberrated image noted in the Discussion. Is that correct? Guiding the reader to the point would again be helpful, especially since they are likely not familiar with the 'correct' morphology.

Yes, that is correct that the AO images show the correct morphology of the hollow structure and the ER of the hypha. The hollow structure is presumed to be a vacuole (Fig. 9b) or a nucleus (Fig. 9c), which should have no fluorescence in this fungal strain. The AO correction shows the improvement of the SNR and image sharpness, which clearly reveals the exclusion of the ER from the hollow structure in all three dimensions. We modified the main text to describe what to look at in Fig. 9b and 9c.

And we have changed the text on page 14 from

The effect of the direct wavefront sensing AO method is evident in the improvement of the SNR, image sharpness and structural definition, as we can see from the comparison of 3D-SIM images before and after sample aberration corrections in Fig. 9b and 9c.

to

The effect of the direct wavefront sensing AO method is evident in the improvement of the SNR, image sharpness and structural definition, showing the correct morphology of the hollow structure and the ER of the hypha. The hollow structure is presumed to be a vacuole (Fig. 9b) or a nucleus (Fig. 9c), which should have no fluorescence in this fungal strain. as we can see from the comparison of 3D-SIM images before and after, sample aberration correction clearly reveals the exclusion of the GFP-labelled ER from the hollow structure in all three dimensions (in Fig. 9b and 9c).

-Table 1 is missing standard deviations and N, please report these so the numbers are statistically meaningful.

We have replaced Table 1 with averages and standard deviations for 5 beads. The data is now displayed as bar graphs. The data and images for individual beads are in Figure 4, Supplementary Figure S2, and Supplementary Table 1.

-Are different lateral/axial scales used in Fig. 4d, and what are the scales in question?

We have added scale bar in both x and z directions. Now Fig. 4c.

Reviewer #2 (Remarks to the Author):

I have received and studied the detailed and comprehensive revisions by Lin et al. My concerns were thoughtfully addressed in their rebuttal. In the process, quite large additions were made to the paper, which

makes the study much more complete: it now compares three different modalities for adaptive optics, and their application to 3D SIM. This goes along with additional samples that were imaged. While I applaud the efforts by the authors, the addition of new technologies, ideas and material also necessitates more reviewing. Thus I hope the author's are not discouraged by the concerns raised below, but can see it as an encouragement that their new ideas were considered exciting and thought-provoking.

We thank the reviewer for his/her encouragement. Below we address his/her remaining concerns.

Most importantly, the new confocal laser spot guide star, filtered with a pinhole, appears to be a novel concept, as it breaks with the notion that one needs a nonlinear (two-photon) process to generate a well isolated spot for wavefront sensing. This would be a welcome change. However, some concerns arose for this. It appears that a judicious choice for pinhole 2 must be made, such that it rejects out-of-focus blur, but does not truncate too many aberrations modes. It was not clear how this choice was made, and how it influences the results. Only a dimension was given (500 microns, for which the corresponding Airy unit should be given too).

Please see our response to reviewer 1. We apologize for not having provided the full details concerning the pinhole in the original manuscript.

Also some discussion on how this compares to the published 2 photon guide star work by the Betzig lab should preferably be included. As another difference, previous art scanned and descanned the guidestar, which is claimed to smooth/average the encountered aberrations. Furthermore, I can see a benefit in scanning, namely that it is easier to get a fluorescence signal if labeling is not uniform.

We have a discussion of the relative merits of the two-photon guide-star vs. the confocal guide star in the discussion. While the two-photon guide-star has the advantage of absolute optical sectioning, it adds considerably to the cost. We agree that scanning/descanning the confocal spot onto the wavefront sensor has benefits, and this is in our plans for the future. The following text is in the Discussion section:

Some applications of AO in fluorescence microscopy have further relied on two-photon excitation to create the guide star for wavefront sensing^{21, 22}. While this has been proven to be an effective approach, it substantially adds to the cost and complexity. Here we have demonstrated a system with confocal illumination that only requires an additional beam path for excitation. A potential improvement to our system would be the addition of a galvo-system for scanning and descanning the confocal spot.

Further concerns arose when the confocal laser spot is used for sensorless AO: The spot itself will get aberrated by the sample, and the iteration of the wavefronts on the DM, both on the way into the sample, and on the way back. Are those forward and backward aberrations problematic? One would need to assume that there is no dispersion in the sample, else the forward (laser) and backward (fluorescence) aberrations would be different, and the correction algorithm would not necessarily converge to the desired correction (fluorescence aberrations).

But beyond that, trying to solve the forward and backward problem at once was always treated with some restraint/concern in the field. Thus, I would like to hear the authors' thoughts on this.

Off note, in the Direct wavefront sensing scheme, it is in an approximation still a single pass measurement. The WFS sensor has such low magnification on each element that some aberrations on the spot will not matter. In my view there it should not be a problem that the ingoing beam is getting aberrated as well (DM + sample).

Aberration correction has been combined previously with confocal microscopy in a configuration where the deformable mirror corrects both the excitation and emission paths (Applied Optics 47(6) pp. 732 – 736, 2008; PNAS 99(9) pp. 5788-5792 2002; Proc. SPIE 8948 894802-1 2014; Opt. Lett. 36(7) pp. 1062 – 1064 2011; Opt Expr. 20(14) pp. 15969 – 15982 2012). Indeed, for confocal imaging, this is the recommended configuration by M. Booth (Proc. SPIE 8948 894802-1 2014). A sensorless AO approach in which the sensorless AO seeks to maximize the peak intensity from the confocal spot will also naturally correct both the excitation and emission. The peak intensity will increase as both are corrected.

If we consider optimizing the wavefront based on the peak intensity from the confocal spot, this configuration will provide a stronger response than widefield illumination because the peak intensity will be enhanced by both the improved excitation and detection PSFs. This is a result of the fact that the confocal psf (for a sufficiently small pinhole) is $\sim h(r)^2$ where $h(r)$ is the widefield PSF. See Booth et al. PNAS 99(9), pp. 5788-5792, 2002.

Because the deformable mirror corrects by adjusting path length, not phase, the discrepancy in wavefront correction between the excitation and emission is due entirely to the refractive index dispersion between the excitation and emission wavelengths. If we estimate the Δn between the excitation 488 and emission 510 to be $2e-3$ (from a dispersion formula for skin from *Tissue Optics* by Valery Tuchin, 2nd ed. SPIE 2007, Eqn. 2.43, p. 252), the accumulated phase error per micron of correction is $2\pi \left(\frac{1}{\lambda_{ex}} + \frac{1}{\lambda_{em}} \right) \frac{\Delta n}{2} \approx 0.025 \text{ radians}$. This is smaller than the residual error after correction which is ~ 0.1 radians.

A question for the Sensorless AO implementation (without confocal guide star): How was the sample illuminated? With a single SIM pattern, a rapid sequence of SIM patterns, or WF?

The sample was illuminated with widefield illumination by applying a blank pattern to the SLM.

We have added the following sentence to the sensorless AO section in the methods:

For sensorless AO, the sample is illuminated either with widefield illumination by turning all SLM pixels on or with the confocal illumination path.

I found it interesting that the sensorless AO performed better under severe aberrations than direct wavefront sensing. I would have assumed that sensorless AO, when buried deep in aberrations, would struggle to dig itself out. I would assume one can design a Shack-Hartmann sensor in many ways, could it be that the one of the authors was designed towards small magnitudes of aberration modes (and as such was more prone to large changes)?

The problem lies in the fact that the DM and sensor are conjugate to one plane and the aberrations are distributed. The situation is illustrated by the cartoon below. The aberrations in an intermediate plane, cause a slight focusing of the emission, so that the beam that reaches the back pupil plane (conjugate to the DM and SHWFS) no longer fills the pupil. In this case the image on the Shack-Hartmann sensor will be smaller than the reference. In the cases shown in Supplemental Figure S5, there is an intensity band running through the back pupil that is missing. This could be due to strong refraction at the edge of the worm body which is something that we have seen in the past (See, for example, Fig. 6b in Thomas et al., J. Biomed. Opt. 2015 Vol. 20 Issue 2 Pages 026006). We have added a discussion to supplemental figure S6 and added the simulation results shown below.

On the biological images (details below), I have for some data difficulties appreciating improved resolution after AO correction. Some line profiles show finer bumps after applications of AO, others the opposite. But line profiles can also be fickle and be prone to picking a suitable one.

Could the authors apply Fourier Ring Correlation or image decorrelation* (which can work on single image frames in contrast to FRC) to assess resolution on biological samples? In terms of resolution gain, the beads show a clear gain, and there quantitative values are given, which was appreciated.

*Descoux, A., Kristin Stefanie Großmayer, and Aleksandra Radenovic. "Parameter-free image resolution estimation based on decorrelation analysis." Nature methods 16.9 (2019): 918-924.

In supplemental tables 4 and 5, we now give some quantitative values for other figures. AO consistently improves the resolution of the structures we have imaged but the degree of improvement varies. In Figure S4, we quantify the lateral and axial width of the interneuron in Fig. 6 before and after sample correction.

Please see our response to reviewer 1 where we try to clarify the improvement in the images after AO. In particular, we note that the comparison of images with and without AO show a very clear improvement in the bead images in Fig. 4 and the images of the endoplasmic reticulum in Fig. 9. The endoplasmic reticulum is a thin membrane. In the AO corrected images it shows up clearly as a thin structure in both lateral and axial cross-sections. In the non-AO corrected images, it appears filled-in in the axial cross-section. The increase in intensity and decrease of linewidth with AO are important indications that the OTF has been improved. There is no wavefront error that will improve the OTF response. Further, we show the OTFs before and after AO correction and the OTF shows increased strength at high frequencies after AO correction. While we agree that the benefits of AO are not always dramatic, we have taken care to show examples where we are certain that the wavefront has been corrected.

We have tried the method of Descloux et al., using their ImageJ plug-in. As the authors note, the method does not work with images that have been deconvolved, including SIM images processed using the standard SIM reconstruction methods (Gustafsson et al., Biophys. J. 94 4957 2008). They discuss this in Supplementary Notes sec. 4 and Supplementary Results sec. 5. We now evaluate the resolution of widefield images with and without AO using their method. This data is provided in the Supplementary Table 3. As you can see there is an increase in resolution with AO except for the case of the worm axon where the resolution value after AO is inexplicably high. We appreciate the suggestion to use their interesting method but we note that it is not infallible.

Further questions:

What is the purpose of the red laser? Is it used to calibrate SWHS sensor and mirror?

Yes, the laser path indicated in red used for DM calibration. The laser is in fact a green diode laser. We have added these details to the figure caption.

We have added the following to the Figure 2 caption:

The blue represents the excitation path (488nm); the green path is the emission; and the red path is used to calibrate the DM.

In the methods section we already had a section describing the red laser path:

To measure the DM actuator influence functions and monitor the wavefront applied to the DM, we added a separate laser beam (520 nm - PL203 Compact Laser Module, Thorlabs) to mimic the emission beam with a flat wavefront. The collimated laser beam enters the optical path after the lens L8, by switching the rotatable mirror *c* to its position 2. The laser beam follows the same path as the emission light beam to the SHWFS.

Figure 8g: is the inset showing the spot before or after correction?

The inset shows the spot after correction. We have now made the image larger.

Figure 8b-c: The resolution does not seem to get higher upon visual impression, but the signal goes up. I cannot see finer features. A bit a similar comment about the data shown in Figure 6-7. It is hard to see finer features, but signal went up. Can the authors please comment on this (or to estimate the resolution with a metric like Image Decorrelation)?

The line profiles show an increase in finer features – there is more structure in the lateral profile (fig 8f) after correction. The lateral image after correction has a more pronounced void in the center.

We have changed the text on the top of page 13 from

By comparing the 3D-SIM images before (Fig. 8b) and after (Fig. 8c) the sample aberration correction in all three dimensions, we can see clearly that fine structural details become sharper and better resolved.

to

To demonstrate the direct wavefront sensing method, we imaged bundles of AJs inside the posterior bulb, as shown in Fig. 8a. In the 3D-SIM image after sample correction, Fig. 8c, these bundles appear sharper.

The insets in Figure 9 B and C, do they show the same axial cross sections? They look structurally quite different. Also in Figure 9c, system correction shows a prominent structure (looking like a diagonally running bar) that is gone in the sample correction. Can the authors please comment on this?

Figures 9 B and C show different structures but the before and after AO axial cross-sections in Figures 9B and 9C are of the same location respectively. The AO correction can have a dramatic effect on the axial resolution. The bar structure in figure 9C before correction does not appear after correction because it no longer appears in that plane, but it is still in the 3D image.

Overall, I think this is important work, it demonstrates a comparison between different AO schemes applied to 3D SIM for the first time and it might introduce a more affordable concept for guide stars.

Thank you.

Sincerely,
Reto Fiolka

Reviewers' Comments:

Reviewer #1:

Remarks to the Author:

I thank the authors for their thorough responses, which have addressed my remaining concerns.

Reviewer #2:

Remarks to the Author:

The authors have carefully addressed my concerns and questions, and have expanded the manuscript with important information. I think an interested reader will find detailed information about the three different AO methods discussed within this manuscript.

As for the concerns about higher resolution in the AO 3D SIM images: I guess when looking at the images (e.g. Figure 5c and 5e) some artifacts in the SIM reconstruction with only "system correction" could be mistaken as higher resolution features? Maybe this could be strengthened instead of just saying the image looks fuzzier without the full AO correction. At first, I mistook some of these fuzzy structures as features. In Figure 9C, it is more apparent to me that the fuzzy structures in the "system correction image" are likely artifacts, whereas with full correction, there reconstruction is basically free of reconstruction artifacts.

Otherwise I am fine with the manuscript, I think it is a great resource for the field of adaptive optics, and may offer some details that have not been given in previous publications.

April 21, 2021

Manuscript NCOMMS-19-1472556B, "Subcellular three-dimensional imaging deep through multicellular thick samples by structured illumination microscopy and adaptive optics" by Lin et al

Response to Reviewer Comments

We thank the reviewers for their comments and appreciate their positive response to our last revision of the manuscript. Below, we address the remaining concern. Their comments are in black, our responses are in blue, and changes to the text are in red.

Reviewer #1 (Remarks to the Author):

I thank the authors for their thorough responses, which have addressed my remaining concerns.

Thank you.

Reviewer #2 (Remarks to the Author):

The authors have carefully addressed my concerns and questions, and have expanded the manuscript with important information. I think an interested reader will find detailed information about the three different AO methods discussed within this manuscript.

As for the concerns about higher resolution in the AO 3D SIM images: I guess when looking at the images (e.g. Figure 5c and 5e) some artifacts in the SIM reconstruction with only "system correction" could be mistaken as higher resolution features? Maybe this could be strengthened instead of just saying the image looks fuzzier without the full AO correction. At first, I mistook some of these fuzzy structures as features. In Figure 9C, it is more apparent to me that the fuzzy structures in the "system correction image" are likely artifacts, whereas with full correction, there reconstruction is basically free of reconstruction artifacts.

We thank the reviewer for pointing this out. Yes, "fuzzy" is not a great way of explaining the difference. We have now changed the sentence from

Fig.5e clearly shows smoother and more continuous perinuclear ER (arrowheads) and peripheral ER (arrows), which look fuzzy in Fig. 5c.

To

Fig.5e clearly shows smoother and more continuous perinuclear ER (arrowheads) and peripheral ER (arrows), which appear wider in Fig. 5c and Fig. 5g.

Otherwise I am fine with the manuscript, I think it is a great resource for the field of adaptive optics, and may offer some details that have not been given in previous publications.

Thank you!

Reto Fiolka